# VADER: VIDEO DIFFUSION ALIGNMENT VIA REWARD GRADIENTS

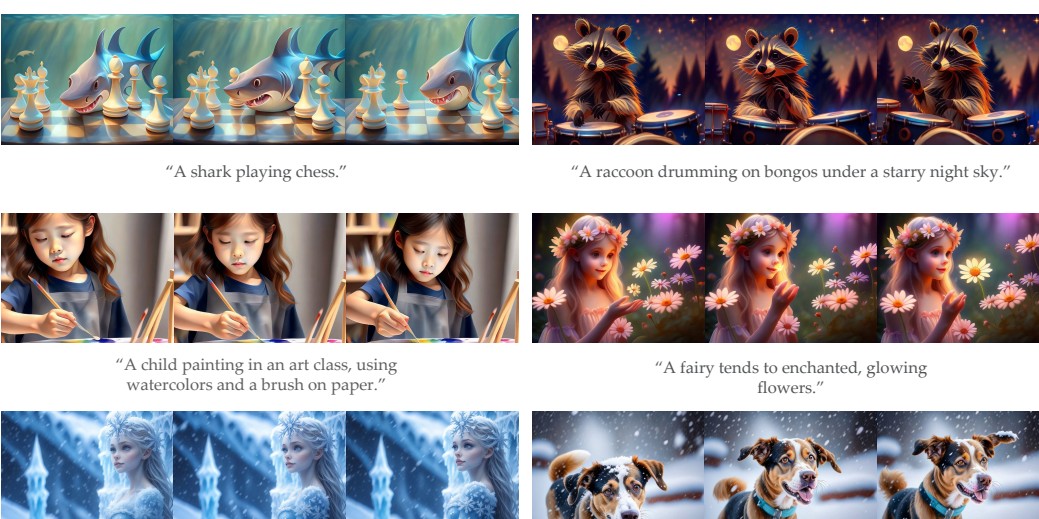

Figure 1: Generations from video diffusion models after adaptation with VADER using reward functions for aesthetics and text-image alignment.

## ABSTRACT

We have made significant progress towards building foundational video diffusion models. As these models are trained using large-scale unsupervised data, it has become crucial to adapt these models to specific downstream tasks. Adapting these models via supervised fine-tuning requires collecting target datasets of videos, which is challenging and tedious. In this work, we utilize pre-trained reward models that are learned via preferences on top of powerful vision discriminative models to adapt video diffusion models. These models contain dense gradient information with respect to generated RGB pixels, which is critical to efficient learning in complex search spaces, such as videos. We show that backpropagating gradients from these reward models to a video diffusion model can allow for compute and sample efficient alignment. We show results across a variety of reward models and video diffusion models, demonstrating that our approach can learn much more efficiently in terms of reward queries and computation than prior gradient-free approaches. More visualization are available at `https://vader-anonymous.github.io/`

## 1 INTRODUCTION

We would like to build systems capable of generating videos for a wide array of applications, ranging from movie production, creative story-boarding, on-demand entertainment, AR/VR content generation, and planning for robotics. The most common current approach involves training foundational video diffusion models on extensive web-scale datasets. However, this strategy, while crucial, mainly

produces videos that resemble typical online content, featuring dull colors, suboptimal camera angles, and inadequate alignment between text and video content.

Contrast this with the needs of an animator who wishes to bring a storyboard to life based on a script and a few preliminary sketches. Such creators are looking for output that not only adheres closely to the provided text but also maintains temporal consistency and showcases desirable camera perspectives. Relying on general-purpose generative models may not suffice to meet these specific requirements. This discrepancy stems from the fact that large-scale diffusion models are generally trained on a broad spectrum of internet videos, which does not guarantee their efficacy for particular applications. Training these models to maximize likelihood across a vast dataset does not necessarily translate to high-quality performance for specialized tasks. Moreover, the internet is a mixed bag when it comes to content quality, and models trained to maximize likelihood might inadvertently replicate lower-quality aspects of the data. This leads us to the question: How can we tailor diffusion models to produce videos that excel in task-specific objectives, ensuring they are well-aligned with the desired outcomes?

The conventional approach to aligning generative models in the language and image domains begins with supervised fine-tuning (Rafailov et al., 2024; Brooks et al., 2023). This involves collecting a target dataset that contains expected behaviors, followed by fine-tuning the generative model on this dataset. Applying this strategy to video generation, however, presents a significantly greater challenge. It requires obtaining a dataset of target videos, a task that is not only more costly and laborious than similar endeavors in language or image domains, but also significantly more complex. Furthermore, even if we were able to collect a video target dataset, the process would have to be repeated for every new video task, making it prohibitively expensive. Is there a different source of signal we can use for aligning video diffusion, instead of collecting a target dataset of desired videos?

Reward models play a crucial role in aligning image and text generations (Schuhmann, 2022; Wu et al., 2023; Lambert et al., 2024). These models are generally built on top of powerful image or text discriminative models such as CLIP or BERT (Radford et al., 2021; Bardes et al., 2023; Tong et al., 2022). To use them as reward models, people either fine-tune them via small amounts of human preferences data (Schuhmann, 2022) or use them directly without any fine-tuning; for instance, CLIP can be used to improve image-text alignment or object detectors can be used to remove or add objects in the images (Prabhudesai et al., 2023).

This begs the question, how should reward models be used to adapt the generation pipeline of diffusion models? There are two broad categories of approaches, those that utilize reward gradients (Prabhudesai et al., 2023; Clark et al., 2023; Xu et al., 2023), and others that use the reward only as a scalar feedback and instead rely on estimated policy gradients (Black et al., 2023; Lee et al., 2023). It has been previously found that utilizing the reward gradient directly to update the model can be much more efficient in terms of the number of reward queries, since the reward gradient contains rich information of how the reward function is affected by the diffusion generation (Prabhudesai et al., 2023; Clark et al., 2023). However, in text-to-image generation space, reward gradient-free approaches are still dominant (Sauer et al., 2024), since these methods can be easily trained within 24 hours and the efficiency gains of leveraging reward gradients are not significant.

In this work, we find that as we increase the dimensionality of generation i.e transition from image to video, the gap between the reward gradient and policy gradient based approaches increases. This is because of the additional amount and increased specificity of feedback that is backpropagated to the model. For reward gradient based approaches, the feedback gradients linearly scale with respect to the generated resolution, as it yields distinct scalar feedback for each spatial and temporal dimension. In contrast, policy gradient methods receive a single scalar feedback for the entire video output. We test this hypothesis as shown in Figure 3, where we find that the gap between reward gradient and policy gradient approaches increases as we increase the generated video resolution. We believe this makes it crucial to backpropagate reward gradient information for video diffusion alignment.

We propose VADER, an approach to adapt foundational video diffusion models using the gradients of reward models. VADER aligns various video diffusion models using a broad range of pre-trained vision models. Specifically, we show results of aligning text-to-video (VideoCrafter, OpenSora, and ModelScope) and image-to-video (Stable Video Diffusion) diffusion models, while using reward models that were trained on tasks such as image aesthetics, image-text alignment, object detection, video-action-classification, and video masked autoencoding. Further, we suggest various tricks to

improve memory usage which allow us to train VADER on a single GPU with 16GB of VRAM. We include qualitative visualizations that show VADER significantly improves upon the base model generations across various tasks. We also show that VADER achieves much higher performance than alternative alignment methods that do not utilize reward gradients, such as DPO or DDPO. Finally, we show that alignment using VADER can easily generalize to prompts that were not seen during training.

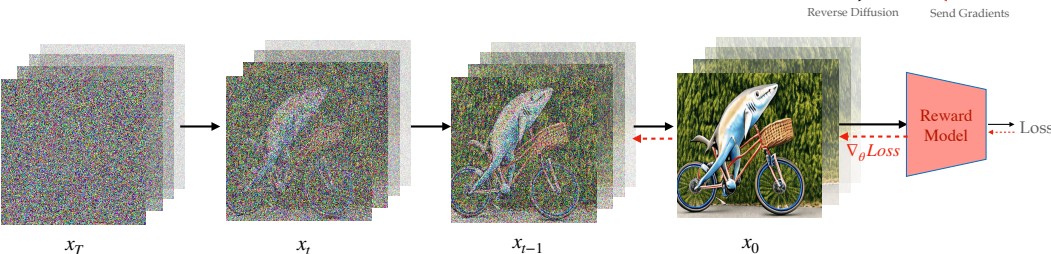

Figure 2: VADER aligns various pre-trained video diffusion models by backpropagating gradients from the reward model, to efficiently adapt to specific tasks.

**Algorithm 1** VADER

**Require:** Diffusion Model weights $\theta$
**Require:** Reward function $R(\cdot)$
**Require:** Denoising Scheduler $f$
        (eg - DDIM, EDM)
**Require:** Gradient cutoff step K
 1: **while** training **do**
 2:    **for** t = T,..,1 **do**
 3:        pred = $\epsilon_\theta(x_t, c, t)$
 4:        **if** t > K **then**
 5:           pred = stop_grad(pred)
 6:        **end if**
 7:        $x_{t-1} = f.\text{step(pred, t, } x_t)$
 8:    **end for**
 9:    $g = \nabla_\theta R(x_0, c)$
10:    $\theta \leftarrow \theta - \eta * g$
11: **end while**

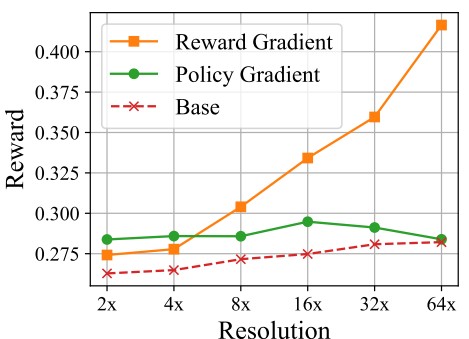

Figure 3: Reward obtained vs resolution of generated video for different methods. We report the reward achieved after 100 steps of optimization. As the resolution of the generation increases, the reward gap between VADER and DDPO significantly increases.

## 2 RELATED WORK

Denoising diffusion models (Sohl-Dickstein et al., 2015; Ho et al., 2020) have made significant progress in generative capabilities across various modalities such as images, videos and 3D shapes (Ho et al., 2022a;b; Liu et al., 2023). These models are trained using large-scale unsupervised or weakly supervised datasets. This form of training results in them having capabilities that are very general; however, most end use-cases of these models have specific requirements, such as high-fidelity generation (Schuhmann, 2022) or better text alignment (Wu et al., 2023).

To be suitable for these use-cases, models are often fine-tuned using likelihood (Blattmann et al., 2023; Brooks et al., 2023) or reward-based objectives (Black et al., 2023; Prabhudesai et al., 2023; Clark et al., 2023; Xu et al., 2023; Lee et al., 2023; Dong et al., 2023; Feng et al., 2023). Likelihood objectives are often difficult to scale, as they require access to the preferred behaviour datasets. Reward or preference based datasets on the other hand are much easier to collect as they require a human to simply provide preference or reward for the data generated by the generative model. Further, widely available pre-trained vision models can also be used as reward models, thus making it much easier to do reward fine-tuning (Black et al., 2023; Prabhudesai et al., 2023). The standard approach

for reward or preference based fine-tuning is to do reinforcement learning via policy gradients (Black et al., 2023; Wallace et al., 2023). For instance, the work of Lee et al. (2023) does reward-weighted likelihood and the work of Black et al. (2023) applies PPO (Schulman et al., 2017). Recent works of Prabhudesai et al. (2023) and Clark et al. (2023) find that instead of using policy gradients, directly backpropagating gradients from the reward model to diffusion process helps significantly with sample efficiency.

A recent method, DPO (Rafailov et al., 2024; Wallace et al., 2023), does not train an explicit reward model but instead directly optimizes on the human preference data. While this makes the pipeline much simpler, it doesn't solve the sample inefficiency issue of policy gradient methods, as it still backpropagates a single scalar feedback for the entire video output.

While we have made significant progress in aligning image diffusion models, this has remained challenging for video diffusion models (Blattmann et al., 2023; Wang et al., 2023). In this work, we take up this challenging task. We find that naively applying prior techniques of image alignment (Prabhudesai et al., 2023; Clark et al., 2023) to video diffusion can result in significant memory overheads. Further, we demonstrate how widely available image or video discriminative models can be used to align video diffusion models. Concurrent to our work, InstructVideo (Yuan et al., 2023) also aligns video diffusion models via human preference; however, this method requires access to a dataset of videos. Such a dataset is difficult to obtain for each different task, and becomes difficult to scale especially to large numbers of tasks. In this work, we show that one can easily align video diffusion models using pre-trained reward models while not assuming access to any video dataset.

## 3 BACKGROUND

Diffusion models have emerged as a powerful paradigm in generative modeling. These models operate by modeling a data distribution through a sequential process of adding and removing noise.

The forward diffusion process transforms a data sample $x$ into a completely noised state over a series of steps $T$. This process is defined by the following equation:

$$x_t = \sqrt{\bar{\alpha}_t}x + \sqrt{1 - \bar{\alpha}_t}\epsilon, \quad \epsilon \sim \mathcal{N}(\mathbf{0}, \mathbf{1}), \tag{1}$$

where $\epsilon$ represents noise drawn from a standard Gaussian distribution. Here, $\bar{\alpha}_t = \prod_{i=1}^{t} \alpha_i$ denotes the cumulative product of $\alpha_i = 1 - \beta_i$, which indicates the proportion of the original data's signal retained at each timestep $t$.

The reverse diffusion process reconstructs the original data sample from its noised version by progressively denoising it through a learned model. This model is represented by $\epsilon_\theta(x_t; t)$ and estimates the noise $\epsilon$ added at each timestep $t$.

Diffusion models can easily be extended for conditional generation. This is achieved by adding $c$ as an input to the denoising model:

$$\mathcal{L}_{\text{diff}}(\theta; \mathcal{D}') = \frac{1}{|\mathcal{D}'|} \sum_{x^i, c^i \in \mathcal{D}'} ||\epsilon_\theta(\sqrt{\bar{\alpha}_t}x^i + \sqrt{1 - \bar{\alpha}_t}\epsilon, c^i, t) - \epsilon||^2, \tag{2}$$

where $\mathcal{D}'$ denotes a dataset consisting of image-conditiong pairs. This loss function minimizes the distance between the estimated noise and the actual noise, and aligns with the variational lower bound for $\log p(x|c)$.

To sample from the learned distribution $p_\theta(x|c)$, one starts with a noise sample $x_T \sim \mathcal{N}(\mathbf{0}, \mathbf{1})$ and iteratively applies the reverse diffusion process:

$$x_{t-1} = \frac{1}{\sqrt{\alpha_t}}\left(x_t - \frac{\beta_t}{\sqrt{1 - \bar{\alpha}_t}}\epsilon_\theta(x_t, t, c)\right) + \sigma_t \mathbf{z}, \quad \mathbf{z} \sim \mathcal{N}(\mathbf{0}, \mathbf{1}), \tag{3}$$

The above formulation captures the essence of diffusion models, which highlights their ability to generate structured data from random noise.

## 4 VADER: Video Diffusion via Reward Gradients

We present our approach for adapting video diffusion models to perform a specific task specified via a reward function $R(\cdot)$.

Given a video diffusion model $p_\theta(\cdot)$, dataset of contexts $D_c$, and a reward function $R(\cdot)$, we seek to maximize the following objective:

$$J(\theta) = \mathbb{E}_{c \sim D_c, x_0 \sim p_\theta(x_0|c)}[R(x_0, c)] \tag{4}$$

To learn efficiently, both in terms of the number of reward queries and compute time, we seek to utilize the gradient structure of the reward function, with respect to the weights $\theta$ of the diffusion model. This is applicable to all reward functions that are differentiable in nature. We compute the gradient $\nabla_\theta R(x_0, c)$ of these differentiable rewards, and use it to update the diffusion model weights $\theta$. The gradient is given by :

$$\nabla_\theta R(x_0, c) = \sum_{t=1}^{T} \frac{\partial R(x_0, c)}{\partial f_t} \cdot \frac{\partial f_t}{\partial \theta}. \tag{5}$$

where $f_t$ is a denoising function that predicts the previous timestep: $x_{t-1} = f(x_t, \theta)$ derived from Equation 3. VADER is flexible in terms of the denoising schedule, we demonstrate results with DDIM (Song et al., 2022) and EDM solver (Karras et al., 2022). To prevent over-optimization, we utilize truncated backpropagation (Tallec & Ollivier, 2017; Prabhudesai et al., 2023; Clark et al., 2023), where the gradient is back propagated only for K steps, where K < T, and T is the total diffusion timesteps. Using a smaller value of K also reduces the memory burden of having to backpropagate gradients, making training more feasible. We provide the pseudocode of the full training process in Algorithm 1. Next, we discuss the type of reward functions we consider for aligning video models.

**Reward Models:** Consider a diffusion model that takes conditioning vector $c$ as input and generates a video $x_0$ of length $N$, consisting of a series of images $i_k$, for each timestep $k$ from 0 to $N$. Then the objective function we maximize is as follows:

$$J_\theta = \mathbb{E}_{c, i_{0:N}}\left[R([i_0, i_1...i_k...i_{N-1}], c)\right] \tag{6}$$

We use a broad range of reward functions for aligning video diffusion models. Below we list down the distinct types of reward functions we consider.

*Image-Text Similarity Reward* - The generations from the diffusion model correspond to the text provided by the user as input. To ensure that the video is aligned with the text provided, we can define a reward that measures the similarity between the generated video and the provided text. To take advantage of popular, large-scale image-text models such as CLIP (Radford et al., 2021), we can take the following approach. For the entire video to be well aligned, each of the individual frames of the video likely need to have high similarity with the context $c$. Given an image-context similarity model $g_{\text{img}}$, we have:

$$R([i_0, i_1...i_k...i_{N-1}], c) = \sum_k R(i_k, c) = \sum_k g_{\text{img}}(i_k, c) \tag{7}$$

Then, we have $J_\theta = \mathbb{E}_{k \in [0,N]}\left[g_{\text{img}}(i_k, c)\right]$, using linearity of expectation as in the image-alignment case. We conduct experiments using the HPS v2 (Wu et al., 2023) and PickScore (Kirstain et al., 2023) reward models for image-text alignment. As the above objective only sits on individual images, it could potentially result in a collapse, where the predicted images are the exact same or temporally incoherent. However, we don't find this to happen empirically, we think the initial pre-training sufficiently regularizes the fine-tuning process to prevent such cases.

*Video-Text Similarity Reward* - Instead of using per image similarity model $g_{\text{img}}$, it could be beneficial to evaluate the similarity between the whole video and the text. This would allow the model to generate videos where certain frames deviate from the context, allowing for richer, more diverse expressive generations. This also allows generating videos with more motion and movement, which

is better captured by multiple frames. Given a video-text similarity model $g_{\text{vid}}$ we have $J_\theta = \mathbb{E}\left[g_{\text{vid}}([i_0, i_1...i_k...i_{N-1}], c)\right]$. In our experiments, we use a VideoMAE (Tong et al., 2022) fine-tuned on action classification, as $g_{\text{vid}}$, which can classify an input video into one of a set of action text descriptions. We provide the target class text as input to the text-to-video diffusion model, and use the predicted probability of the ground truth class from VideoMAE as the reward.

*Image Generation Objective* - While text similarity is a strong signal to optimize, some use cases might be better addressed by reward models that only sit on the generated image. There is a prevalence of powerful image-based discriminative models such as Object Detectors and Depth Predictors. These models utilize the image as input to produce various useful metrics of the image, which can be used as a reward. The generated video is likely to be better aligned with the task if the reward obtained on each of the generated frames is high. Hence we define the reward in this case to be the mean of the rewards evaluated on each of the individual frames, i.e $R([i_0, i_1...i_k...i_{N-1}], c) = \sum_k R(i_k)$. Note that given the generated frames, this is independent of the text input $c$. Hence we have, $J_\theta = \mathbb{E}_{k \in [0,N]}[R(i_k)] = \mathbb{E}_{k \in [0,N]}[M_\phi(i_k)]$ via linearity of expectation, where $M_\phi$ is a discriminative model that takes an image as input to produce a metric, that can be used to define a reward. We use the Aesthetic Reward model (Schuhmann, 2022) and Object Detector (Fang et al., 2021) reward model for our experiments.

*Video Generation Objective* - With access to an external model that takes in multiple image frames, we can directly optimize for desired qualities of the generated video. Given a video metric model $N_\phi$, the corresponding reward is $J_\theta = \mathbb{E}\left[N_\phi([i_0, i_1, ..i_k...i_{N-1}])\right]$.

VideoCrafter                    VADER (Ours)

"The raccoon is wearing a red coat and holding a snowball."

"The fox is wearing a red hat and playing with leaves."

"A strong lion and a graceful lioness resting together in the shade of a big tree on a wide grassland."

"A peaceful deer eating grass in a thick forest, with sunlight filtering through the trees."

Figure 4: Text-to-video generation results for VideoCrafter and VADER. We show results for VideoCrafter Text-to-Video model on the left and results for VADER on the right, where we use VideoCrafter as our base model. The reward models applied are a combination of HPSV2.1 and Aesthetic model in the first two rows, and PickScore in the last two rows.

*Long-horizon consistent generation* - In our experiments, we adopt this formulation to enable a feature that is quite challenging for many open-source video diffusion models - that of generating clips that are longer in length. For this task, we use Stable Video Diffusion (Blattmann et al., 2023), which is an image-to-video diffusion model. We increase the context length of Stable Video Diffusion by 3x by making it autoregressive. Specifically, we pass the last generated frame by the model as input for generating the next video sequence. However, we find this to not work well, as the model

was never trained over its own generations thus resulting in a distribution shift. In order to improve the generations, we use a video metric model $N_\phi$ (V-JEPA (Bardes et al., 2023)) that given a set of frames, produces a score about how predictive the frames are from one another. We apply this model on the autoregressive generations, to encourage these to remain consistent with the earlier frames. Training the model in this manner allows us to make the video clips temporally and spatially coherent.

**Reducing Memory Overhead:** Training video diffusion models is very memory intensive, as the amount of memory linearly scales with respect to the number of generated frames. While VADER significantly improves the sample efficiency of fine-tuning these models, it comes at the cost of increased memory. This is because the differentiable reward is computed on the generated frame, which is a result of sequential de-noising steps.

*Standard Tricks* - To reduce the memory usage we use LoRA (Hu et al., 2021) that only updates a subset of the model parameters, further we use mixed precision that stores non-trainable parameters in fp16. Also, gradient checkpointing is applied. For the long horizon tasks, offload the storage of the backward computation graph from the GPU memory to the CPU memory.

*Truncated Backprop* - Additionally, In our experiments we only backpropagate through the diffusion model for one timestep, instead of backpropagating through multiple timesteps (Prabhudesai et al., 2023), and have found this approach to obtain competitive results while requiring much less memory.

*Subsampling Frames* - Since all the video diffusion models we consider are latent diffusion models, we further reduce memory usage by not decoding all the frames to RGB pixels. Instead, we randomly subsample the frames and only decode and apply loss on the subsampled ones.

We conduct our experiments on 2 A6000 GPUS (48GB VRAM), and our model takes an average of 12 hours to train. However, our codebase supports training on a single GPU with 16GB VRAM.

## 5 RESULTS

In this work, we focus on fine-tuning various conditional video diffusion models, including VideoCrafter (Chen et al., 2024) , Open-Sora (Zheng et al., 2024) , Stable Video Diffusion (Blattmann et al., 2023) and ModelScope (Wang et al., 2023), through a comprehensive set of reward models tailored for images and videos. These include the Aesthetic model for images (Schuhmann, 2022), HPSv2 (Wu et al., 2023) and PickScore (Kirstain et al., 2023) for image-text alignment, YOLOS (Fang et al., 2021) for object removal, VideoMAE for action classification (Tong et al., 2022), and V-JEPA (Bardes et al., 2023) self-supervised loss for temporal consistency. Our experiments aim to answer the following questions:

- How does VADER compare against gradient-free techniques such as DDPO or DPO regarding sample efficiency and computational demand?
- To what extent can the model generalize to prompts that are not seen during training?
- How do the fine-tuned models compare against one another, as judged by human evaluators?
- How does VADER perform across a variety of image and video reward models?

This evaluation framework assesses the effectiveness of VADER in creating high-quality, aligned video content from a range of input conditioning.

**Baselines.** We compare VADER against the following methods:

- **VideoCrafter** (Chen et al., 2024), **Open-Sora 1.2** (Zheng et al., 2024), and **ModelScope** (Wang et al., 2023) are current state-of-the-art (publicly available) text-to-video diffusion models. We serve them as base models for fine-tuning in text-to-video space.
- **Stable Video Diffusion** (Blattmann et al., 2023) is the current state-of-art publicly image-to-video diffusion model. We use it for all our experiments in image-to-video space.
- **DDPO** (Black et al., 2023) is a recent image diffusion alignment method that uses policy gradients to adapt diffusion model weights. Specifically, it applies PPO algorithm (Schulman et al., 2017) to the diffusion denoising process. We extend it to adapt video diffusion models.

- **Diffusion-DPO** (Wallace et al., 2023) extends the recent development of Direct Preference Optimization (DPO) (Rafailov et al., 2024) in the LLM space to image diffusion models. They show that directly modeling the likelihood using the preference data can alleviate the need for a reward model. We extend their implementation to align video diffusion models, where we use the reward model to obtain the required preference data.

**Reward models.**    We use the following reward models to fine-tune the video diffusion model.

- **Aesthetic Reward Model**: We use the LAION aesthetic predictor V2 (Schuhmann, 2022), which takes an image as input and outputs its aesthetic score in the range of 1-10. The model is trained on top of CLIP image embeddings.
- **Human Preference Reward Models**: We use HPSv2 (Wu et al., 2023) and PickScore (Kirstain et al., 2023), which take as input an image-text pair and predict human preference.
- **Object Removal**: We design a reward model based on YOLOS (Fang et al., 2021), an object detection model, from which a video model learns to remove the target object category.
- **Video Action Classification**: We employ a reward model based on VideoMAE (Tong et al., 2022). Our reward is the probability predicted by the action classifier given a video as input.
- **Temporal Consistency via V-JEPA**: We also use V-JEPA (Bardes et al., 2023) as our reward model to improve temporal consistency, where the reward is the negative of the masked autoencoding loss in the V-JEPA feature space.

**Prompts.**    We consider various prompt datasets for reward fine-tuning of text-to-video and image-to-video diffusion models. For more details, please refer to subsection A.1.

## 5.1    SAMPLE AND COMPUTATIONAL EFFICIENCY

Training of large-scale video diffusion models is done by a small set of entities with access to a large amount of computing; however, fine-tuning of these models is done by a large set of entities with access to a small amount of computing. Thus, it becomes imperative to have fine-tuning approaches that boost both sample and computational efficiency.

In this section, we compare VADER's sample and compute efficiency with other reinforcement learning methods such as DDPO and DPO. In Figure 5, we visualize the reward curves during training, where the x-axis in the upper half of the figure is the number of reward queries and the one in the bottom half is the GPU-hours. As can be seen, VADER is significantly more sample and compute efficient than DDPO or DPO. This is mainly due to the fact that we send dense gradients from the reward model to the diffusion weights, while the baselines only backpropagate scalar feedback.

## 5.2    GENERALIZATION ABILITY

Table 1: Reward on prompts in train & test. Base model is ModelScope in this experiment. We split the prompts into train and test sets, such that the prompts in the test set do not have any overlap with the ones for training. We find that VADER achieves the best on both metrics.

| Method | Aes (T2V) | | HPS (T2V) | | ActP | Aes (I2V) | |
| --- | --- | --- | --- | --- | --- | --- | --- |
| | Train. | Test. | Train. | Test. | Train. | Train. | Test. |
| Base | 4.61 | 4.49 | 0.25 | 0.24 | 0.14 | 4.91 | 4.96 |
| DDPO | 4.63 | 4.52 | 0.24 | 0.23 | 0.21 | N/A | N/A |
| DPO | 4.71 | 4.41 | 0.25 | 0.24 | 0.23 | N/A | N/A |
| Ours | **7.31** | **7.12** | **0.33** | **0.32** | **0.79** | **7.83** | **7.64** |

A desired property of fine-tuning is generalization, i.e. the model fine-tuned on a limited set of prompts has the ability to generalize to unseen prompts. In this section, we extensively evaluate this property across multiple reward models and baselines. While training text-to-video (T2V) models, we use HPSv2 Action Prompts in our training set, whereas we use Activity Prompts in our test set. We use Labrador dog category in our training set for training image-to-video (I2V) models, while Maltese category forms our test set. Table 1 showcases VADER's generalization ability.

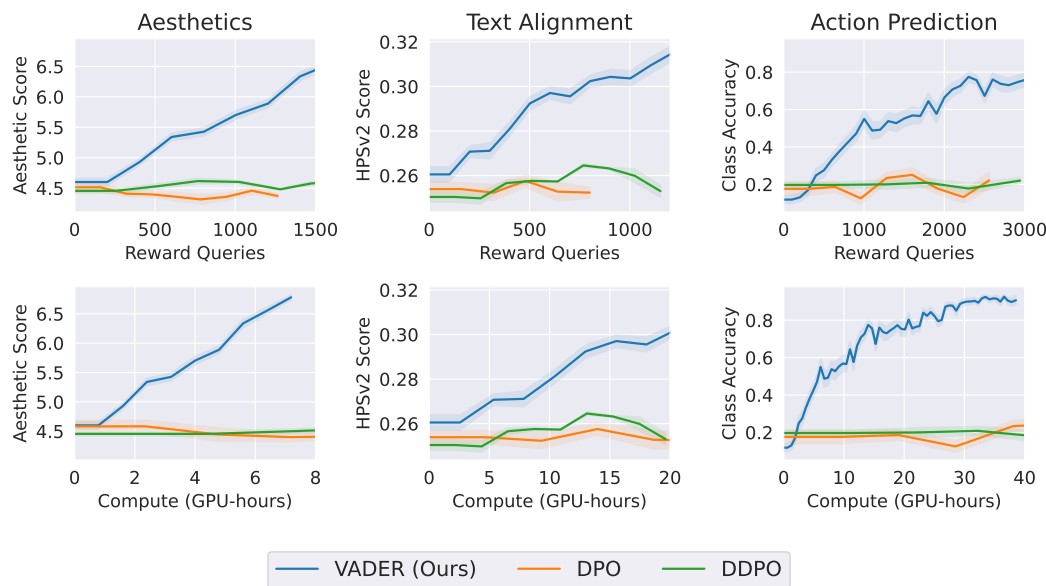

Figure 5: Training efficiency comparison. **Top**: Sample efficiency comparison with DPO and DDPO. **Bottom**: Computational efficiency comparison with DPO and DDPO.

## 5.3 HUMAN EVALUATION

We carried out a study to evaluate human preferences via Amazon Mechanical Turk. The test consisted of a side-by-side comparison between VADER and ModelScope. To test how well the videos sampled from both the models aligned with their text prompts, we showed participants two videos generated by both VADER and a baseline method, asking them to identify which video better matched the given text. For evaluating video quality, we asked participants to compare two videos generated in response to the

Table 2: Human Evaluation results for HPS reward model, where the task is image-text alignment.

| Method | Fidelity | Text Align |
|---|---|---|
| ModelScope | 21.0% | 39.0% |
| VADER (**Ours**) | **79.0%** | **61.0%** |

same prompt, one from VADER and one from a baseline, and decide which video's quality seemed higher. We gathered 100 responses for each comparison. The results, illustrated in Table 2, show a preference for VADER over the baseline methods.

## 5.4 QUALITATIVE VISUALIZATION

In this section, we visualize the generated videos for VADER and the respective baseline. We conduct extensive visualizations across all the considered reward functions on various base models.

**HPS Reward Model:** In Figure 4, we visualize the results before and after fine-tuning VideoCrafter using both HPSv2.1 and Aesthetic reward function in the top two rows. Before fine-tuning, the raccoon does not hold a snowball, and the fox wears no hat, misaligning with the text; however, the videos generated from VADER do not result in these inconsistencies. Further, VADER successfully generalizes to unseen prompts as shown in the first row of Figure 8, where the dog's paw is less like a human hand than the video on the left. Similar improvements can be observed in videos generated from Open-Sora V1.2 and ModelScope as shown in the second and third rows of Figure 7.

**Aesthetic Reward Model:** In Figure 4, in the top two rows we visualize the results before and after fine-tuning VideoCrafter using a combination of Aesthetic reward function and HPSv2.1 model. Also, we fine-tune ModelScope via Aesthetic Reward function and demonstrate its generated video in the last row of Figure 7. We observe that fine-tuning makes the generated videos more artistic.

Before        VADER (Ours)

"A book and a cup of tea on a blanket in a sunflower field."

"A book and a cup of hot chocolate on a windowsill with a snowy view."

"A book and a cup of coffee on a rustic wooden table in a cabin."

Figure 6: Object removal using VADER. **Left:** Base model (VideoCrafter) generations, **Right:** VADER generations after fine-tuning to not display books using an object detector as a reward model. VADER effectively removes book and replaces it with some other object.

**PickScore Model:** In the bottom two rows of Figure 4, we showcase videos generated by PickScore fine-tuned VideoCrafter. VADER shows improved text-video alignment than the base model. In Figure 8, we test both models using a prompt that is not seen during training time. Further, video generated from PickScore fine-tuned Open-Sora is displayed in Figure 7.

**Object Removal:** Figure 6 displays the videos generated by VideoCrafter after fine-tuning using YOLOS-based objection removal reward function. In this example, books are the target objects for removal. These videos demonstrate the successful replacement of books with alternative objects.

**Video Action Classification:** In Figure 10, we visualize the video generation of ModelScope and VADER. In this case, we fine-tune VADER using the action classification objective, for the action specified in the prompt. For "A person eating donuts", we find that VADER makes the human face more evident along with adding sprinkles to the donut. Earlier generations are often misclassified as baking cookies, which is a different action class in the kinetics dataset. The addition of colors and sprinkles to the donut makes it more distinguishable from cookies leading to a higher reward.

**V-JEPA reward model:** In Figure 9, we show results for increasing the length of the video generated by Stable Video Diffusion (SVD). For generating long-range videos on SVD, we use autoregressive inference, where the last frame generated by SVD is given as conditioning input for generating the next set of images. We perform three steps of inference, thus expanding the context length of SVD by three times. However, as one can see in the images bordered in red, after one step of inference, SVD starts accumulating errors in its predictions. This results in deforming the teddy bear, or affecting the rocket in motion. VADER uses V-JEPA objective of masked encoding to enforce self-consistency in the generated video. This manages to resolve the temporal and spatial discrepancy in the generations as shown in Figure 9.

## 6 CONCLUSION

We presented VADER, which is a sample and compute efficient framework for fine-tuning pre-trained video diffusion models via reward gradients. We utilized various reward functions evaluated on images or videos to fine-tune the video diffusion model. We further showcased that our framework is agnostic to conditioning and can work on both text-to-video and image-to-video diffusion models. We hope our work creates more interest towards adapting video diffusion models.

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

## A  APPENDIX

### A.1  PROMPTS.

We consider the following set of prompt datasets for reward fine-tuning of text-to-video and image-to-video diffusion models.

- **Activity Prompts (Text):**  We consider the activity prompts from the DDPO (Black et al., 2023). Each prompt is structured as "a(n) [animal] [activity]," using a collection of 45 familiar animals. The activity for each prompt is selected from a trio of options: "riding a bike", "playing chess", and "washing dishes".

- **HPSv2 Action Prompts (Text):**  Here we filter out 50 prompts from a set of prompts introduced in the HPS v2 dataset for text-image alignment. We filter prompts such that they contain action or motion information in them.

- **ChatGPT Created Prompts (Text):**  We prompt ChatGPT to generate some vivid and creatively designed text descriptions for various scenarios, such as books placed beside cups, animals dressed in clothing, and animals playing musical instruments.

- **ImageNet Dog Category (Image):**  For image-to-video diffusion model, we consider the images in the Labrador retriever and Maltese category of ImageNet as our set of prompts.

- **Stable Diffusion Images (Image):**  Here we consider all 25 images from Stable Diffusion online demo webpage.

### A.2  VISUALIZATIONS.

Before                                                    VADER (Ours)

Open-Sora

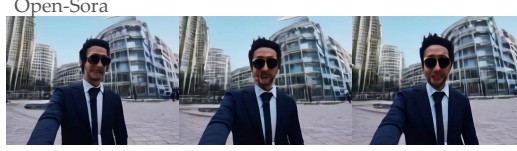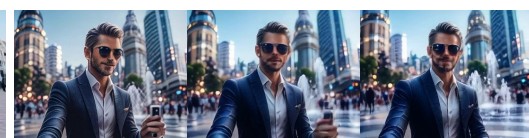

"a man in a trendy suit taking a selfie in a city square, surrounded by modern buildings and a fountain."

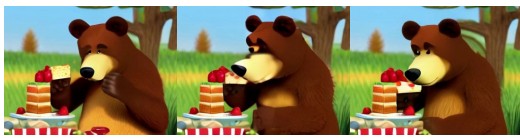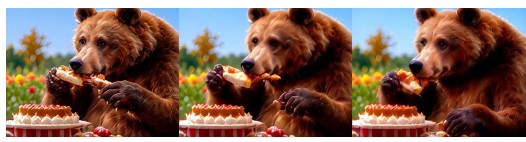

"A bear enjoying a slice of cake at a picnic."

ModelScope

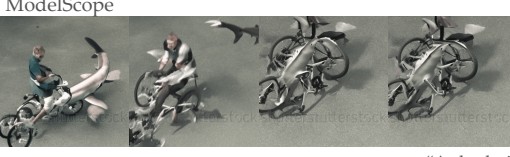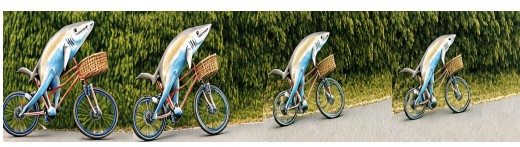

"A shark riding a bike."

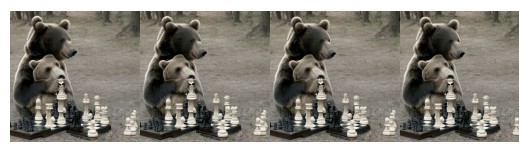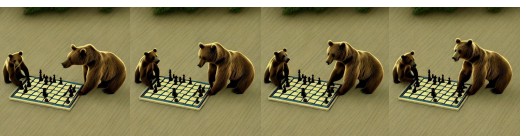

"A bear playing chess."

Figure 7: Aligning Open-Sora 1.2 and ModelScope with VADER. The left column shows results from the base models, while results from VADER are demonstrated on the right. The first two rows use Open-Sora as the base model, and the last two rows use ModelScope. The reward models applied are PickScore in the first row, HPSv2.1 in the second row, HPSv2 in the third row, and the Aesthetic reward model in the last row.

VideoCrafter                                    VADER (Ours)

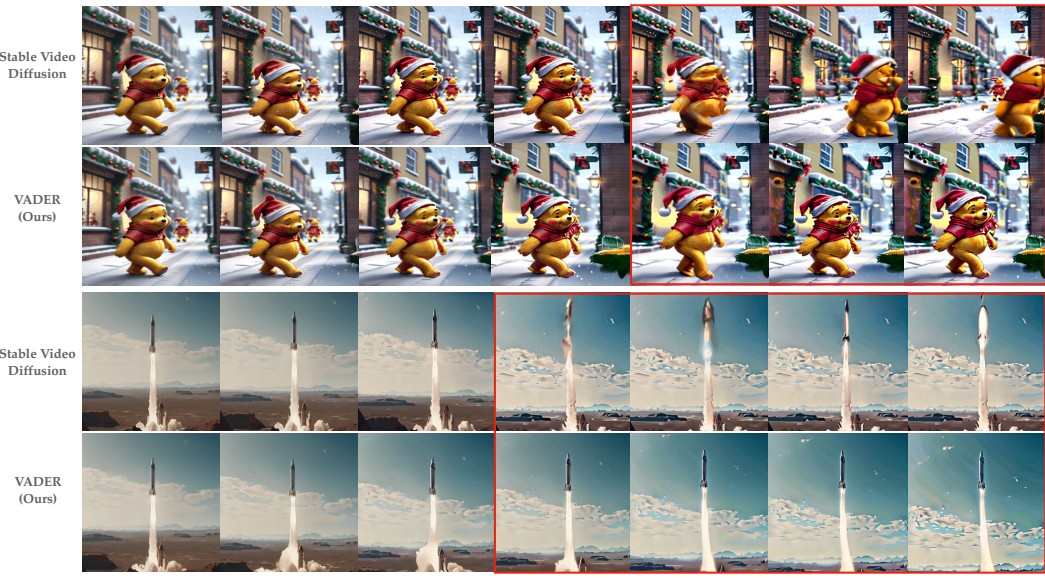

"A dog playing a slide guitar on a porch during a gentle rainstorm."

"A dog strumming an acoustic guitar by a lakeside campfire under the stars."

Figure 8: Additional text-to-video generation results for VideoCrafter and VADER. We show results for VideoCrafter Text-to-Video model on the left and results for VADER on the right, where we use VideoCrafter as our base model. The reward models applied are a combination of HPSV2.1 and Aesthetic model in the first row, and PickScore in the last row. The videos are generated based on prompts that are not encountered during training.

Stable Video Diffusion

VADER (Ours)

Stable Video Diffusion

VADER (Ours)

Figure 9: Improving temporal and spatial consistency of Stable Video Diffusion (SVD) Image-to-Video Model. Given the leftmost frame as input, we use autoregressive inference to generate 3*N frames in the future, where N is the context length of SVD. However, this suffers from error accumulation, resulting in corrupted frames, as highlighted in the red border. We find that VADER can improve the spatio-temporal consistency of SVD by using V-JEPA's masked encoding loss as its reward function.

"A person eating Donuts"

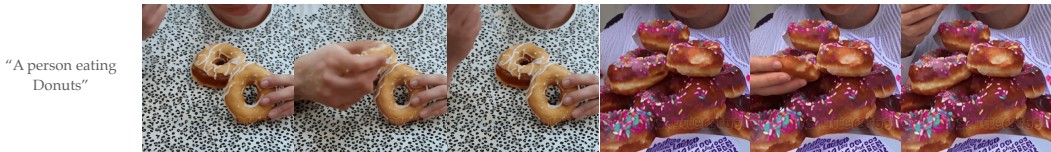

Figure 10: Video action classifiers as reward model. We use VideoMAE action classification model as a reward function to fine-tune ModelScope's Text-to-Video Model.

### A.3 BASELINES

**DDPO**: We implement the approach in (Black et al., 2023), by closely following the code from kvablack/ddpo-pytorch. We adopt the same code structure, which computes the log probability for each ddim denoising step, for the video diffusion model. This log probability is used to compute policy gradient, following the method in (Black et al., 2023). We use the PPO version of the method as opposed to only using policy gradient, since this is reported to give slightly better performance. We use the same LoRA and gradient checkpointing approach that is used in VADER for updating the video diffusion model.

**DiffusionDPO**: Our implementation for the approach in (Wallace et al., 2023) builds on the code from SalesforceAIResearch/DiffusionDPO. We alternate between sampling from the diffusion model, and training the model via the DPO objective, similar to the process in DDPO. Samples are added to a replay buffer (since offline samples can be used for DPO training), and we use batches sampled from the buffer for training. Given pairs of video generations, we assign them as $V^w$ and $V^l$ based on rewards from the reward model, where $V^w$ obtains higher reward. We then use the exact same loss function as them. We set $\beta$, the KL penalty in DPO to be 5000 following the standard. Just as for DDPO, the model is updated using same LoRA training and gradient checkpointing approach as in VADER.

### A.4 QUANTATIVE ANALYSIS.

In this section, unless explicitly specified otherwise, the base model used is VideoCrafter. We use the following acronyms to refer to various methods in the Tables below.

- **VADER-Pick**: PickScore reward.
- **VADER-HPS**: Aesthetic and HPS rewards
- **VADER-ViCLIP**: ViCLIP and Aesthetic rewards.
- **VADER-V-JEPA**: V-JEPA reward.
- **VideoCrafter**, **ModelScope**, and **Stable Video Diffusion**: Baseline models.

In this section, we present a comprehensive analysis of VADER. In Table 3, we explore how optimizing VADER with specific reward models affects other reward scores. Notably, fine-tuning with the PickScore reward model significantly improves the HPS score, which indicates a strong inter-reward correlation. This aligns with the findings in Figure 11.

In Table 4, we evaluate VADER variants using EvalCrafter (Liu et al., 2024) benchmark. We find that all VADER variants outperform the base model in both temporal coherence and motion quality. We further evaluate VADER on VBench benchmark (Huang et al., 2024) in Table 5. We find that VADER-Pick achieves the best consistency scores for subjects and backgrounds, while VADER-HPS excels in aesthetic and imaging quality. Further, Table 6 evaluates VADER-V-JEPA and Stable Video Diffusion using VBench metrics, demonstrating significant improvements in temporal coherence and aesthetic style.

Table 7 ablates various memory optimization techniques. We find that using certain techniques, the total memory usage can be reduced from 276.3 GB to 32.5 GB. In Table 8 we study the diversity of generation across models. In Table 9 we ablate the number of truncated backpropagation steps, showing how different values of the backpropagation step ($K$) in the diffusion model influence training results. Finally, in Figure 12, we compare VADER's compute efficiency against standard and on-policy versions of DDPO and DPO baselines when trained for longer.

Table 3: In this Table, we study how optimizing for specific reward functions via VADER affects scores on other reward functions. We observe that the HPS score increases significantly after fine-tuning the base model via the PickScore model, indicating a strong correlation.

| Model | HPS Score | PickScore Score | Aesthetic Score | ViCLIP Score |
|---|---|---|---|---|
| VideoCrafter | 0.2564 | 20.9231 | 5.2219 | **0.2643** |
| VADER-HPS | 0.2651 | 21.1345 | **5.7965** | 0.2622 |
| VADER-Pick | **0.2669** | **21.4911** | 5.5757 | 0.2640 |
| VADER-ViCLIP | 0.2511 | 20.8927 | 5.6241 | 0.2628 |

Table 4: EvalCrafter (Liu et al., 2024) evaluation results for VADER. EvalCrafter calculates **Temporal Coherence** using Warping Error, Semantic Consistency (cosine similarity of the embeddings of consecutive frames), and Face Consistency, which assess frame-wise pixel and semantic consistency. **Motion Quality** is evaluated through Action-Score (action classification accuracy), Flow-Score (average optical flow between frames obtained from RAFT (Teed & Deng, 2020)), and Motion AC-Score (amplitude classification consistency with the text prompt). We generate 700 videos from each model for this comparision. Results demonstrate that all the VADER-variants outperform the base model (VideoCrafter).

| Model | Temporal Coherence | Motion Quality |
|---|---|---|
| VideoCrafter | 55.90 | 52.89 |
| VADER-HPS | 59.65 | **55.46** |
| VADER-Pick | **60.75** | 54.65 |
| VADER-ViCLIP | 57.08 | 54.25 |

Table 5: VBench (Huang et al., 2024) evaluation results VADER. The metrics used in VBench include: **Subject Consistency** (consistency of the main subject across frames, evaluated using DINO (Caron et al., 2021) feature similarity), **Background Consistency** (using CLIP (Radford et al., 2021) feature similarity), **Motion Smoothness** (fluidity of motion, based on motion priors from a frame interpolation model), **Dynamic Degree** (extent of motion in the video, estimated with RAFT), **Aesthetic Quality** (assessed via the LAION aesthetic predictor), and **Imaging Quality** (using MUSIQ (Ke et al., 2021)). We generate 700 videos for each model using prompts not seen during training. We find that VADER-Pick has the best consistency score, while VADER-HPS shows the best aesthetic and imaging quality.

| Model | Subject Consistency | Background Consistency | Motion Smoothness | Dynamic Degree | Aesthetic Quality | Imaging Quality |
|---|---|---|---|---|---|---|
| VideoCrafter | 0.9544 | 0.9652 | 0.9688 | 0.5346 | 0.5752 | 0.6677 |
| VADER-HPS | 0.9659 | 0.9713 | **0.9734** | 0.4741 | **0.6295** | **0.7145** |
| VADER-Pick | **0.9668** | **0.9727** | 0.9726 | 0.3732 | 0.6094 | 0.6762 |
| VADER-ViCLIP | 0.9564 | 0.9662 | 0.9714 | **0.5519** | 0.6008 | 0.6566 |

Table 6: VBench evaluation results for Image to Video diffusion models. The base model is Stable Video Diffusion. We compare Stable Video Diffusion and VADER-V-JEPA. VADER-V-JEPA demonstrates improvements across most metrics, particularly in consistency and aesthetic quality.

| Model | Subject Consistency | Background Consistency | Motion Smoothness | Dynamic Degree | Aesthetic Quality | Imaging Quality |
|---|---|---|---|---|---|---|
| Stable Video Diffusion | 0.9042 | 0.9469 | 0.9634 | 0.8333 | 0.6782 | 0.6228 |
| VADER-V-JEPA | **0.9401** | **0.9551** | **0.9669** | 0.8333 | **0.6807** | **0.6384** |

Table 7: Ablation of memory usage for different components in ModelScope-based VADER. For this experiment, we offload the memory to the CPU main memory to prevent GPU out-of-memory error. Starting with the standard (**LoRA + Mixed Precision**), each row represents an added component (**Subsampling Frames**, **Truncated Backpropagation**, and **Gradient Checkpointing**) applied incrementally to the previous row. The total RAM reduced is about 240 GB after implementing all the steps.

| Method | VRAM | System RAM | Total RAM |
|---|---|---|---|
| LoRA + Mixed Precision | 12.1 GB | 264.2 GB | 276.3 GB |
| + Subsampling Frames | 12.1 GB | 216.8 GB | 228.9 GB |
| + Truncated Backpropagation | 12.1 GB | 57.3 GB | 69.4 GB |
| + Gradient Checkpointing | 12.1 GB | 20.4 GB | **32.5 GB** |

Table 8: Diversity of generated videos for VADER. We generate 500 videos for each model and prompt combination. We use 5 prompts resulting in a total of 2500 videos per model. The diversity is calculated using the variance of VideoMAE latent space embeddings across the 500 videos for each prompt. We then average the variances over all prompts. We find that VADER variants exhibit reduced diversity compared to the baseline model (**VideoCrafter**). Prior works (Kirk et al., 2023; Murthy et al., 2024) have found similar results, where aligning a model for a specific use case often results in reduced diversity.

| | VideoCrafter | VADER-Pick | VADER-HPS | VADER-ViCLIP |
|---|---|---|---|---|
| **Average Variance** | **0.0037** | 0.0026 | 0.0023 | 0.0031 |

Table 9: We ablate the number of truncated backpropagation steps ($K$) in VADER. For this experiment, we use VADER trained using Aesthetic and HPS Rewards. We find that higher values of $K$ result in more semantic level changes, while $K = 1$ results in more fine-grained changes, specifically in the earlier steps of training. Visualizations are available at Project Website. Further, we find as we train longer, both the models start exhibiting semantic level changes. We also find it is easier to optimize with a smaller value of $K$, as can be seen in the results below.

| Training Step | Reward Value ($K$=1) | Reward Value ($K$=10) |
|---|---|---|
| 1 | 5.047 | 5.0946 |
| 100 | 5.3342 | 5.2523 |
| 200 | 5.4977 | 5.2072 |
| 300 | **5.6479** | 5.1906 |

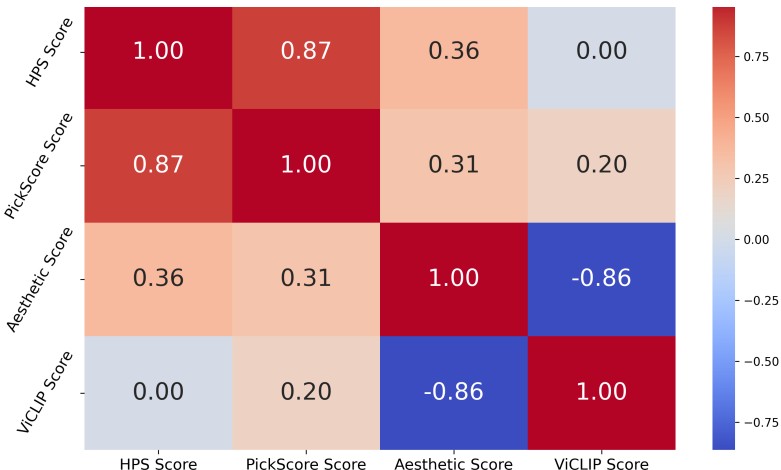

Figure 11: We study correlation across different reward models for VADER. We find that there is a strong positive correlation between PickScore and HPS scores, while a strong negative correlation between ViCLIP and Aesthetic reward function.

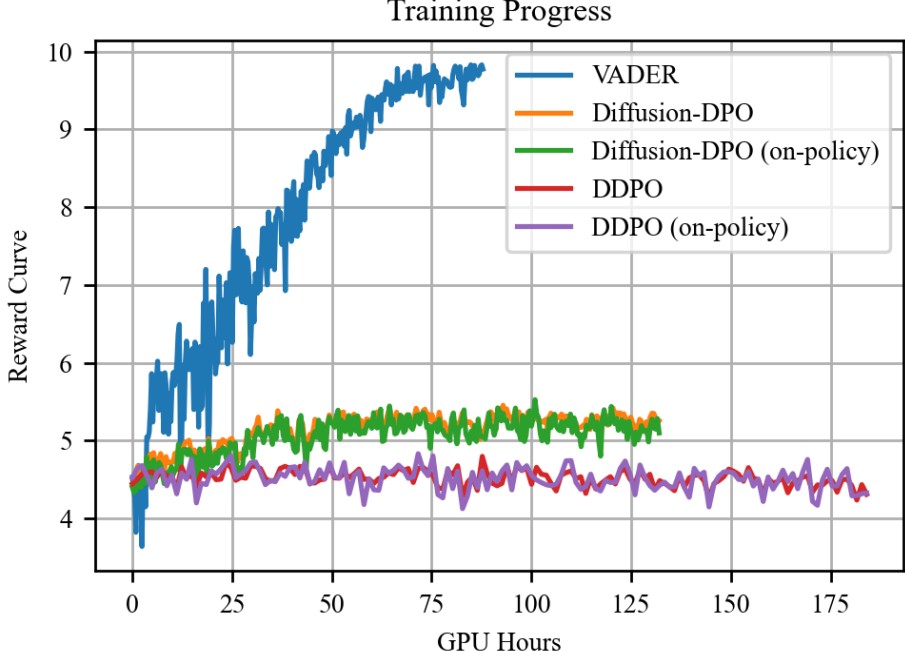

Figure 12: Training efficiency comparison against various baselines, when trained for longer. The base model used here is ModelScope. We compare VADER against DPO, DDPO, on-policy DPO, and on-policy DDPO. For implementing the on-policy version of the baselines, we simply reduce the UTD (update-to-data) ratio to 1, thus only doing a single gradient update for each datapoint sampled. We observe that VADER significantly outperforms all of them in terms of compute efficiency.

