# OpenReview forum: "VADER: Video Diffusion Alignment via Reward Gradients"
_ICLR.cc/2025/Conference — Submitted to ICLR 2025_

### Official Review · Reviewer_tG9V · 2024-11-03

**Soundness:** 4
**Presentation:** 4
**Contribution:** 4
**Rating:** 8
**Confidence:** 4

**Summary:**

This work introduces VADER, a method for aligning video diffusion models using reward gradients. VADER repurposes off-the-shelf vision discriminative models as reward models to adapt video diffusion models more effectively. Additionally, the paper presents practical techniques to optimize memory usage, enabling efficient training of VADER on standard hardware.

**Strengths:**

Originality:
VADER is a novel approach to aligning video diffusion models using reward gradients rather than policy gradient methods. It creatively repurposes various pre-trained vision models as reward models, expanding their utility in video generation.

Quality:
The paper is methodologically rigorous, with strong experimental results demonstrating clear performance gains. Additionally, memory-optimizing techniques make VADER more accessible, broadening its potential user base.

Clarity:
The work is well-organized, with clear explanations and visualizations that effectively showcase the benefits of reward gradients and diverse reward models for alignment.

Significance:
VADER significantly advances practical video generation, making it more accessible and adaptable. This positions VADER as a valuable contribution to generative AI in video synthesis.

**Weaknesses:**

The paper lacks a quantitative evaluation of the temporal coherence achieved with the v-jepa reward model. While Figure 9 provides qualitative evidence of improvement, the analysis would be more robust with a quantitative assessment of temporal consistency. Adding such an experiment would offer a more comprehensive understanding of VADER's performance in maintaining coherence over time when trained on the v-jepa reward.

While VADER incorporates multiple reward models, the paper lacks a detailed examination of how each specific reward model influences alignment objectives like temporal coherence, aesthetic quality, or text-video alignment. Additionally, it would be valuable to understand how optimizing for one reward, such as aesthetics, might impact the performance on other metrics, like temporal coherence.

**Questions:**

How are the visualizations selected?
Are the visual examples in the paper randomly chosen, or were they curated to highlight specific successes of VADER? Understanding the selection process would clarify whether the results are representative or potentially cherry-picked.

Why aren’t popular metrics, such as FID/FVD, used in the experiments?
Frechet Video Distance (FVD) is a commonly used metric for evaluating video quality in generative models, albeit with its own limitations and pitfalls. Including it would allow for a more standardized and comparable evaluation.

---

> ### Author Response · Authors · 2024-11-28
>
> **Q 6.1: Lacks a quantitative evaluation of the temporal coherence over V-JEPA and other models.**
>
> Thanks for pointing this out, we have added temporal dynamics and temporal coherence evaluation over VADER trained using V-JEPA, ViCLIP, PickScore, HPS and Aesthetics reward functions on our webpage. We use VBench [1] and EvalCrafter [2] for conducting this eval.
>
> VBench Evaluation while using **image to video models*, while trained using **V-Jepa reward** model:
>
> | Model                  | Subject Consistency | Background Consistency | Motion Smoothness | Dynamic Degree | Aesthetic Quality | Imaging Quality |
> |------------------------|---------------------|------------------------|-------------------|----------------|-------------------|-----------------|
> | Stable Video Diffusion | 0.9042             | 0.9469                | 0.9634           | 0.8333         | 0.6782           | 0.6228          |
> | VADER-V-JEPA           | **0.9401**         | **0.9551**            | **0.9669**       | 0.8333         | **0.6807**       | **0.6384**      |
>
>
>
> VBench Evaluation while using **text to video models**, while trained using **various reward models**:
>
>
> | Model                     | Subject Consistency | Background Consistency | Motion Smoothness | Dynamic Degree | Aesthetic Quality | Imaging Quality |
> |---------------------------|----------------------|-------------------------|-------------------|----------------|-------------------|-----------------|
> | VideoCrafter              | 0.9544              | 0.9652                 | 0.9688           | 0.5346         | 0.5752            | 0.6677          |
> | VADER-Aesthetic and HPS   | 0.9659              | 0.9713                 | **0.9734**       | 0.4741         | **0.6295**        | **0.7145**      |
> | VADER-PickScore           | **0.9668**          | **0.9727**             | 0.9726           | 0.3732         | 0.6094            | 0.6762          |
> | VADER-ViCLIP and Aesthetic| 0.9564              | 0.9662                 | 0.9714           | **0.5519**     | 0.6008            | 0.6566          |
>
>
> **EvalCrafter Evaluation** using text-to-video models can be found in the Table below:
>
> | Model                           | Temporal Coherence | Motion Quality |
> |---------------------------------|--------------------|----------------|
> | VideoCrafter                    | 55.90             | 52.89          |
> | VADER-Aesthetic and HPS         | 59.65             | 55.46          |
> | VADER-PickScore                 | **60.75**         | 54.65          |
> | VADER-Aesthetic and ViCLIP      | 57.08             | 54.25          |
>
>
> We also visualize the generated videos after training on each of these reward models in the [project website](https://vader-anonymous.github.io/#aesthetic-and-viclip-reward).
>
> Overall we find that VADER beats the baseline over all these metrics. We find that video reward models such as ViCLIP help achieve higher temporal dynamics, while image-reward models such as PickScore help achieve higher temporal coherence and subject consistency. Lastly, we find that fine-tuning via V-JEPA improves both temporal coherence and motion quality of the original Stable Video Diffusion.
>
> **Q 6.2: Lacks a detailed examination of how each specific reward model influences alignment objectives.**
>
> Thanks for the suggestion. We show a detailed comparison on how finetuning using a specific reward model affects the results on the other reward models: https://vader-anonymous.github.io/#reward-correlation . We find that there is a strong positive correlation between PickScore and HPS reward models because both of them contribute to text-video alignment while there is a somewhat negative correlation between Aesthetic Score and ViCLIP Score. We visualize the correlation matrix on the webpage and in  Figure 11 of the paper.
>
>
> **Q 6.3: How are the visualizations selected?**
> The examples were not selected entirely random. To show an entirely random selection we visualize a lot more videos here:  https://vader-anonymous.github.io/Video_Gallery.html. We have also conducted a human eval study and our results are shown in Table 2.
>
> **Q 6.4 Why aren’t popular metrics, such as FID/FVD, used in the experiments?**
> FVD or FID requires access to a real video dataset to compute the distance between generated and real distributions. VADER is data-free and does not rely on any real video data, so FID or FVD are not applicable.
>
>
> [1] VBench: Comprehensive Benchmark Suite for Video Generative Models
> [2] EvalCrafter: Benchmarking and Evaluating Large Video Generation Models

---

> > ### Comment · Area_Chair_TYiK · 2024-11-30
> >
> > Dear Reviewer,
> >
> > The authors have provided their responses. Could you please review them and share your feedback?
> >
> > Thank you!

---

> ### Comment · Reviewer_tG9V · 2024-11-30
>
> I thank the authors for their response. They have addressed all my questions and concerns. I have no further questions.

---

### Official Review · Reviewer_mSP2 · 2024-11-03

**Soundness:** 3
**Presentation:** 3
**Contribution:** 2
**Rating:** 5
**Confidence:** 4

**Summary:**

This paper introduces VADER, a method for fine-tuning video diffusion models using reward gradients to improve task-specific alignment. By utilizing pre-trained reward models as discriminators, VADER enhances video quality, text alignment, and temporal consistency. The approach employs memory optimization techniques to enable efficient training, even with limited resources. Experimental results demonstrate that VADER achieves strong performance across various video generation tasks.

**Strengths:**

1. The authors propose VADER, which uses reward gradients to fine-tune video diffusion models, achieving efficient task adaptation.
2. This paper experiments with various reward models, making it suitable for multiple video generation tasks and achieving strong results in both subjective evaluations and quantitative metrics.
3. This paper employs several optimization techniques (such as truncated backprop) to enable operation in resource-limited environments.

**Weaknesses:**

1. The proposed VADER approach, which fine-tunes video diffusion models using reward gradients, optimizes network parameters with various reward models serving as discriminators. This enables improved adaptation to specific tasks. However, the use of reward models in generative model training is a well-explored concept. This paper just extends that approach within the context of diffusion models.
2. Since using reward models to backpropagate gradients requires the diffusion model to produce fully denoised outputs, all denoising steps must be executed, which places high demands on training resources. This might also lead to very small batch sizes. Although the authors employ several tricks to reduce resource usage, it raises the question of whether these adjustments impact the training outcomes. For instance, how much does backpropagation through only one timestep in the diffusion model affect the network parameters?
3. Some visual results in the paper still show misalignment between text and image content. For example, in Figure 7, the prompt "A bear playing chess" leads VADER to generate two bears.

**Questions:**

See Weakness

---

> ### Author Response · Authors · 2024-11-28
>
> **Q 5.1 How does K=1 in VADER affect the outcomes?**
>
> We run comparisons with K=10, by offloading the GPU memory to the CPU memory. We find different tradeoffs for using a higher value of K.
>
> In the earlier steps of training, higher values of K results in more semantic level changes, while K=1 results in more fine-grained changes. Further, we find as we train longer, both the models start exhibiting semantic level changes.
>
> We also find, training a model at K=10 is much slower than training it at K=1, for instance with 12 GPU hours, K=10 takes 50 gradient update steps, while K=1 takes 350 gradient update steps. We also find that K=1 is much easier to optimize, for instance with 200 gradient update steps, K=1 gets a reward of 5.49, while K=10 gets a reward of 5.26.
>
> This result is shown here: https://vader-anonymous.github.io/#truncated-backpropagation-ablation
>
> Overall, we think the right value of K, might be system specific, higher values are preferred when the training system is not GPU VRAM bottlenecked.
>
> **Q 5.2 Misalignment between text and image content in Figure 7 in two bears example.**
>
> Figure 7 of the paper was generated by a model fine-tuned using the Aesthetic Reward Model. This reward model is trained to predict the aesthetic quality of pictures, so it does not contribute much to image-text alignment. Fine-tuning by utilizing the HPS or PickScore or ViCLIP Reward Models would help text alignment much more. More visualizations for video-text alignment are available at https://vader-anonymous.github.io/.

---

> > ### Comment · Reviewer_mSP2 · 2024-11-29
> >
> > I thank the authors for their response.
> >
> > I agree with reviewer sDFV and HJ2c regarding their concerns about novelty. Using a discriminator to directly backward the gradient is widely used in many studies and lacks significant novelty. This work seems more like an application of gradient backpropagation in video diffusion models. While the authors' findings in the ablation studies on the number of truncated backpropagation steps are interesting, they are insufficient to support this paper's novelty.
> >
> > Additionally, the authors emphasize their contribution as aligning video diffusion models using reinforcement learning. In my understanding, aligning with textual descriptions should be a crucial aspect of this. The authors noted that models fine-tuned with the Aesthetic Reward Model fail to align with text, because it is trained to predict the aesthetic quality of pictures. However, the results using the PickScore Reward still face this issue. For example, in Figure 8, the output does not depict a starry sky as described. Similarly, in the newly presented experimental results titled “DOODL vs. VADER,” prompts like *2 dogs and a whale, ocean adventure* and *Teddy bear and 3 real bears* continue to exhibit this problem.
> >
> > Therefore, I have decided to give a score of 5.

---

> ### Author Response · Authors · 2024-11-30
>
> **Q5.3: Lack of Textual description alignment**
>
> We thank the reviewer for their prompt response. VADER improves the textual alignment score of VideoCrafter from 0.256 to 0.267 when evaluated via HPS score, when evaluated via human eval for text alignment, VADER's generations are preferred 72\% of the times over the baseline. These numbers are averaged over 700 videos generated by both the model. We therefore think this is a more accurate metric to evaluate the text-alignment of VADER than evaluating specific videos shown in the paper. We never claim VADER has perfect video-text alignment, however we stand with the claim that on average it's considerably better than the videos generated by VideoCrafter or ModelScope in terms of text-alignment.

---

> > ### Author Response · Authors · 2024-12-03
> >
> > Dear Reviewer,
> >
> > As the rebuttal discussion is coming to an end, If there are any experiments we could run, or concerns that we can address that can help you better evaluate our work. Please let us know!
> >
> > Since our last discussion, we have done the following comparisons:
> >
> > **Diversity Test**
> >
> > We visualize the generations from VADER and baseline to see if the diversity gets negatively affected due to it. Please find them here:  https://vader-anonymous.github.io/Video_Diversity.html
> >
> > **Comparision againt T2V-Turbo**
> >
> > We have conducted evaluation against T2V-Turbo, on VBench benchmark.  T2V-Turbo showcases results on this benchmark in their paper. We find that VADER-Aesthetic+HPS outperforms both T2V-Turbo (4-step) and T2V-Turbo (8-step), while we use the same weighted average formula as used in their paper. Note that we never optimize for this benchmark, neither directly nor indirectly. Evaluation on the VBench benchmark was an afterthought after we had trained our models.
> >
> > Please find the results in the Table below or on the following link: https://vader-anonymous.github.io/#vbench-evaluation
> >
> >
> > | Model                          | Subject Consistency | Background Consistency | Motion Smoothness | Dynamic Degree | Aesthetic Quality | Imaging Quality | Weighted Average |
> > |--------------------------------|---------------------|-------------------------|-------------------|----------------|-------------------|-----------------|------------------|
> > | VideoCrafter                   | 0.9544              | 0.9652                  | 0.9688            | 0.5346         | 0.5752            | 0.6677          | 0.7997           |
> > | T2V Turbo (4 Steps)            | 0.9639              | 0.9656                  | 0.9562            | 0.4771         | 0.6183            | 0.7266          | 0.8126           |
> > | T2V Turbo (8 Steps)            | **0.9735**          | **0.9736**              | 0.9572            | 0.3686         | 0.6265            | 0.7168          | 0.8058           |
> > | VADER-Aesthetic and HPS        | 0.9659              | 0.9713                  | **0.9734**        | 0.4741         | **0.6295**        | 0.7145          | **0.8167**       |
> > | VADER-PickScore                | 0.9668              | 0.9727                  | 0.9726            | 0.3732         | 0.6094            | 0.6762          | 0.7971           |
> > | VADER-Aesthetic and ViCLIP     | 0.9564              | 0.9662                  | 0.9714            | **0.5519**     | 0.6008            | 0.6566          | 0.8050           |
> >
> > Thank you.

---

### Official Review · Reviewer_sDFV · 2024-11-04

**Soundness:** 3
**Presentation:** 3
**Contribution:** 2
**Rating:** 6
**Confidence:** 4

**Summary:**

The paper proposed a reward fine-tune method called VADER. By using dense gradient information tied to generated RGB pixels, VADER enables more efficient learning in complex spaces like video generation. Backpropagating gradients from reward models to the video diffusion model facilitates both compute and sample-efficient alignment. Results across various reward and video diffusion models demonstrate that this gradient-based approach learns more efficiently than previous gradient-free methods in terms of reward queries and computational resources.

**Strengths:**

1. The paper proposes to use gradient information for critic model preference tuning.
2. The experiment results are good.

**Weaknesses:**

1. Lack of novelty, the paper is more likely to be a tech report rather than a paper. Directly backward the gradient is not novel for RL.
2. The paper proposes a on-policy strategy, while all the comparisons are off-policy strategies which is not fair. There are so many on-policy strategies[1] that perform better than off-policy strategies.
3. Missing experiment details. How do you use DPO/DDPO on your dataset since they need preference pairs for training. How do you create the preference pair?


[1] Yuan, Weizhe, et al. "Self-Rewarding Language Models." Forty-first International Conference on Machine Learning.

**Questions:**

1. More methods should be considered for comparison.
2. More details of the experiments should be enclosed.

---

> ### Author Response · Authors · 2024-11-28
>
> **Q 4.1: Lack of novelty.**
>
> Please refer to the global answer.
>
> **Q 4.2: Unfair comparison between an on-policy strategy and off-policy strategies.**
>
> We agree that VADER is more on-policy compared to DDPO and DPO. However we do not think this is the reason why VADER is more sample-efficient. In-fact off-policy methods are generally considered to be more sample efficient than on-policy methods [1].
>
> To further investigate this, we created an on-policy version of DPO and DDPO, we do this by taking a single gradient update per video sampled. In our experiments we don’t find the results to improve the sample efficiency of the baselines. We plot these results in the following link: https://vader-anonymous.github.io/#training-efficiency-comparison and in Figure 12 in the main paper. Overall we think VADER is more sample efficient, because it backpropagates dense feedback to the model weights, while DDPO or DPO backpropagate scalar feedback.
>
>
> **Q 4.3: More methods should be considered for comparison.**
>
> There are not many methods in video-diffusion alignment, therefore we adapt state-of-the-art methods in image diffusion alignment space to videos.  Since the deadline we have also added other baselines such as DDPO-on policy, DPO-on policy and DOODL , the results for these methods can be found in https://vader-anonymous.github.io/#training-efficiency-comparison and https://vader-anonymous.github.io/#doodl-vader.
>
>
> **Q 4.4: Experimental details on DPO and DDPO.**
>
> We implement DDPO [2], by closely following the official code from https://github.com/kvablack/ddpo-pytorch. We adopt the same code structure, which computes the log probability for each ddim denoising step, for the video diffusion model. This log probability is used to compute policy gradients, following the method code. We use the PPO version of the method as opposed to only using policy gradient, since this is reported to give slightly better performance in their paper. We use the same LoRA and gradient checkpointing approach that is used in VADER for updating the video diffusion model
>
>
> We implement DiffusionDPO [3] based on the official code from https://github.com/SalesforceAIResearch/DiffusionDPO. We alternate between sampling from the diffusion model, and training the model via the DPO objective, similar to the process in DDPO. Samples are added to a replay buffer (since offline samples can be used for DPO training), and we use batches sampled from the buffer for training. Given pairs of video generations, we assign them as $V^w$ and $V^l$ based on rewards from the reward model, where $V^w$ obtains higher reward. We then use the same
> loss function as the original paper, which increases the likelihood of the $V^w$ sample. We set β, the KL penalty in DPO to be 5000 following the standard. Just like DDPO, the model is updated using same LoRA training and gradient checkpointing approach as in VADER.
>
> Thanks for pointing this out, we have added the explanation in the paper.
>
> -----------
> References:
>
> [1] Efficient Off-Policy Meta-Reinforcement Learning via Probabilistic Context Variables
>
> [2] Training Diffusion Models with Reinforcement Learning
>
> [3] Diffusion Model Alignment Using Direct Preference Optimization
>
> [4] End-to-End Diffusion Latent Optimization Improves Classifier Guidance

---

> > ### Comment · Reviewer_sDFV · 2024-11-29
> > **Some comments**
> >
> > I really appreciate the effort of the author. Unfortunately, to my knowledge, the author's paper is not the first to successfully align video diffusion models using reinforcement learning. The known T2V-Turbo[1] also utilized multiple reward models as feedback, was first released on May 29, and has been accepted by NeurIPS 2024. Although there are differences in implementation, I believe T2V-Turbo surpasses VADER in terms of large-scale experimental design and results. Therefore, I think the novelty of this paper is limited, and I maintain my original score.
> >
> > [1] Li, Jiachen, et al. "T2V-Turbo: Breaking the Quality Bottleneck of Video Consistency Model with Mixed Reward Feedback." arXiv preprint arXiv:2405.18750 (2024).

---

> > ### Comment · Reviewer_sDFV · 2024-11-29
> > **Question about the experiment。**
> >
> > Moreover, I understand that in Table 1, the authors utilized multiple reward models to simultaneously perform gradient backpropagation. However, when comparing with DPO, the authors claimed to use a replay buffer to store preference pairs scored by the reward models. Yet, different reward models might produce varying or even contradictory results for the same dataset. In other words, different reward models could have differing opinions on the winner-loser pairs. I am curious about how the authors handled the generation of offline preference data under the influence of multiple reward models?

---

> ### Author Response · Authors · 2024-11-29
>
> We thank the reviewer for their quick response!
>
> **Q 4.5: Related work - T2V-Turbo**
>
> Thanks for sharing this work! We were not aware of the work of T2V-Turbo [1]. We plan to cite them and have also started working on comparing our results against them.
>
> However, we would like to point out some major differences between the two works:
>
> - The **goal of the T2V-Turbo is very different from goal of VADER**. Their goal is to improve the inference speed of T2V models, by distilling a teacher model (eg: VideoCrafter) into a few-step video consistency model.  The goal of VADER is instead to align existing video-models to various downstream tasks. For instance,  we show results on removing objects from the scene while using object detectors as reward models - https://vader-anonymous.github.io/#object-removal-reward . This could be a very useful downstream task for removing explicit content from video generation pipelines but this doesn't align with the goal of T2V-Turbo, as again their goal is to improve the speed of video generation. Further as both approaches have very different goals, this gives rise to very different evaluations studied in both the approaches, for instance we compare against other forms of RL techniques such as DDPO or DPO and we focus on training efficiency of VADER in terms  of compute and samples used during finetuning. T2V-Turbo on the other hand focuses on inference time compute efficiency, and hasn't reported any numbers studying finetuning efficiency.
>
> - **T2V-Turbo requires having access to an external video-text training dataset**, VADER on the other hand is entirely data-free. Explicit requirement on an external training dataset, could potentially make it difficult to adapt to different base video models, as the base models considered might have never been trained on certain datasets.
>
> - Lastly our approach is not restricted to T2V models and can be easily adapted to I2V mdoels as we show here: https://vader-anonymous.github.io/#v-jepa-reward
>
> Irrespective of the major differences between the two works, we still think it's not fair to consider T2V-Turbo's work, when evaluating the novelty/originality of VADER. We consider T2V-Turbo to be a concurrent work due to the following reasons:
>
> - T2V-Turbo  was arxiv-only and was not released as a NeurIPS paper until the abstract deadline for ICLR (Sept 27th)
> - Our methods and initial results were submitted to conferences and our project webpage was available by end-February, which is way before the arxiving date for T2V-Turbo.  Unfortunately we can't share exact details on this, while preserving the sanctity of double blind reviewing.
>
> Having said that we do acknowledge that T2V-Turbo indeed does RL on video diffusion models. We have therefore **corrected** our previous statement stating that: "we are the first work to successfully align video diffusion models using reinforcement learning".
>
> **Q 4.6: Table 1 Clarification**
>
> Thanks for pointing this out. We believe there is a misunderstanding here, In Table 1 we do not use multiple reward models at once, instead the results are shown while using each reward model in the column **independently** for fine-tuning. We consider each reward model to be a different downstream task, and show how effectively VADER can fit to a variety of downstreams tasks. We hope this clarifies the comparison with DPO. We will include this clarification in the paper.

---

> > ### Comment · Area_Chair_TYiK · 2024-11-30
> >
> > Dear Reviewer,
> >
> > The authors have provided their responses. Could you please review them and share your feedback?
> >
> > Thank you!

---

> ### Comment · Reviewer_sDFV · 2024-11-30
> **Feedback**
>
> Thanks for the clarification in Table 1.
>
> For the related work, first of all, T2V-Turbo has already demonstrated in the ablation study the impact of image model rewards and video model rewards on aspects such as aesthetics, dynamic quality, and motion smoothness, etc. , rather than merely using RL to improve inference speed. Furthermore, one concern I have with this paper is that T2V-Turbo was trained and evaluated using a larger-scale dataset and compared against numerous well-known methods, making its results more reliable than those presented in this paper.
>
> I agree with the authors that this work is a concurrent work with T2V-Turbo. But I think the authors need to carefully review more papers before claiming a very strong argument.
>
> I will raise my score to 5.

---

> > ### Author Response · Authors · 2024-12-02
> >
> > **Comparision against T2V turbo**
> >
> > We have conducted evaluation against T2V-Turbo, on VBench benchmark.  T2V-Turbo showcases results on this benchmark in their paper. We find that VADER-Aesthetic+HPS outperforms both T2V-Turbo (4-step) and T2V-Turbo (8-step), while we use the same weighted average formula as used in their paper. Note that we never optimize for this benchmark, neither directly nor indirectly. Evaluation on the VBench benchmark was an afterthought after we had trained our models.
> >
> > Please find the results in the Table below or on the following link: https://vader-anonymous.github.io/#vbench-evaluation
> >
> >
> > | Model                          | Subject Consistency | Background Consistency | Motion Smoothness | Dynamic Degree | Aesthetic Quality | Imaging Quality | Weighted Average |
> > |--------------------------------|---------------------|-------------------------|-------------------|----------------|-------------------|-----------------|------------------|
> > | VideoCrafter                   | 0.9544              | 0.9652                  | 0.9688            | 0.5346         | 0.5752            | 0.6677          | 0.7997           |
> > | T2V Turbo (4 Steps)            | 0.9639              | 0.9656                  | 0.9562            | 0.4771         | 0.6183            | 0.7266          | 0.8126           |
> > | T2V Turbo (8 Steps)            | **0.9735**          | **0.9736**              | 0.9572            | 0.3686         | 0.6265            | 0.7168          | 0.8058           |
> > | VADER-Aesthetic and HPS        | 0.9659              | 0.9713                  | **0.9734**        | 0.4741         | **0.6295**        | 0.7145          | **0.8167**       |
> > | VADER-PickScore                | 0.9668              | 0.9727                  | 0.9726            | 0.3732         | 0.6094            | 0.6762          | 0.7971           |
> > | VADER-Aesthetic and ViCLIP     | 0.9564              | 0.9662                  | 0.9714            | **0.5519**     | 0.6008            | 0.6566          | 0.8050           |

---

> > > ### Comment · Reviewer_sDFV · 2024-12-02
> > > **Feedback**
> > >
> > > I truly appreciate the authors' efforts and the thorough comparison presented. I hope all the results will be shown in the revised version. Based on this, I am happy to raise my score to 6.

---

> > > ### Comment · Reviewer_sDFV · 2024-12-03
> > > **Some Questions**
> > >
> > > You mentioned, "T2V-Turbo showcases results on this benchmark in their paper," implying that the results from the paper were used as the reference in the table. However, I couldn't find any matching results in the T2V-Turbo paper that align with those reported in the table. Could you clarify which specific table or section in their paper you are referencing?

---

> ### Author Response · Authors · 2024-12-03
>
> Sorry for the confusion, we use the same evaluation pipeline/benchmark as them i.e VBench, which involves the same evaluation metrics such as Subject Consistency,Background Consistency, Motion Smoothness,Dynamic Degree, Aesthetic Quality	and Imaging Quality.
>
> However, To ensure consistency with our Human Eval and VBench results, we use the same set of 700 prompts officially released from the EvalCrafter benchmark (https://github.com/evalcrafter/EvalCrafter/tree/master/prompts, ).  Note the same set of 700 EvalCrafter prompts were also used by T2V-Turbo for human-evaluation in Section 4.2.
>
> Further we also follow the VBench custom prompt pipeline for evaluation: https://github.com/Vchitect/VBench?tab=readme-ov-file#new-evaluate-your-own-videos, and use the officially released checkpoints from T2V-Turbo.
>
> We will mention all of these details in the revised version.

---

> ### Author Response · Authors · 2024-12-03
>
> **Evaluation against T2V-Turbo while using Standard Prompt Suite of VBench**
>
> We conduct an additional experiment while using the standard prompt suite of VBench. For the baselines, we copy-paste the numbers reported by T2V-Turbo in Table 1 of their paper. We follow the standard evaluation pipeline on VBench, for evaluating VADER (https://github.com/Vchitect/VBench?tab=readme-ov-file#evaluation-on-the-standard-prompt-suite-of-vbench).
>
> We find that, VADER achieves the best Quality Score, amongst all the baselines. Quality Score is the weighted sum of the normalized score of each metric, as reported in VBench and T2V-Turbo.
>
> We report the results in the Table below and the following link: https://vader-anonymous.github.io/#vbench-evaluation-standard-prompt-suite.
>
>
> | Model                       | Subject Consistency | Background Consistency | Motion Smoothness | Dynamic Degree | Aesthetic Quality | Imaging Quality | Temporal Flickering | Quality Score |
> |-----------------------------|---------------------|-------------------------|-------------------|----------------|-------------------|-----------------|---------------------|---------------|
> | **VideoCrafter2**           | 96.85              | 98.22                  | 97.73            | 42.50          | 63.13             | 67.22           | 98.41              | 82.20         |
> | **Pika**                    | 96.76              | **98.95**                  | 99.51            | 37.22          | 63.15             | 62.33           | **99.77**              | 82.68         |
> | **Gen-2**                   | **97.61**              | 97.61                  | **99.58**            | 18.89          | 66.96             | 67.42           | 99.56              | 82.47         |
> | **T2V Turbo (VC2)**         | 96.28              | 97.02                  | 97.34            | 49.17          | 63.04             | **72.49**           | 97.48              | 82.57         |
> | **VADER - Aesthetic and HPS** | 95.79              | 96.71                  | 97.06            | **66.94**          | **67.04**             | 69.93           | 98.19              | **84.15**     |
>
>
> **Evaluation against T2V-Turbo on EvalCrafter benchmark**
>
> To further evaluate temporal coherence and motion quality, we use the official Evalcrafter benchmark while using their standard  prompts. We find that VADER outperforms T2V-Turbo on these metrics, as shown in the table below and the following link: https://vader-anonymous.github.io/#evalcrafter-evaluation
>
> | Model                          | Temporal Coherence | Motion Quality |
> |--------------------------------|--------------------|----------------|
> | VideoCrafter2                  | 55.90             | 52.89          |
> | T2V Turbo (4 Steps)            | 57.10             | 54.93          |
> | T2V Turbo (8 Steps)            | 57.05             | 55.34          |
> | VADER-Aesthetic and HPS        | 59.65             | **55.46**          |
> | VADER-PickScore                | **60.75**         | 54.65          |
> | VADER-Aesthetic and ViCLIP     | 57.08             | 54.25          |
>
> We will include these results in the revised version of our paper.

---

### Official Review · Reviewer_wLqi · 2024-11-04

**Soundness:** 3
**Presentation:** 3
**Contribution:** 2
**Rating:** 8
**Confidence:** 4

**Summary:**

The authors introduce an alignment tuning method for video generation models utilizing gradient backpropagation from reward models. This approach addresses a critical need for producing high-quality, aligned video content. By directly guiding the generation model through reward gradients, the method achieves notable sample and computational efficiency compared to gradient-free approaches.

**Strengths:**

- The problem to be solved is clearly defined, and its importance is well-acknowledged.

- The use of reward model gradients is highly intuitive and well-explained. Additionally, as the proposed methodology is data-free, it is practically useful and less likely to inherit biases from datasets used in alignment fine-tuning.

- Experimental results show this method is computationally efficient compared to gradient-free approaches such as DDPO and DPO.

**Weaknesses:**

**Clarity of Contribution:** It is unclear what the authors’ unique contributions are. Utilizing reward model gradients for alignment tuning has already been demonstrated in text-to-image generation by Prabhudesai et al. and Clark et al. The authors should clarify what specific advancements they are claiming in this area.

Additionally, the authors mention that there are significant memory overhead issues in video models when implementing these methods, yet a clear, step-by-step ablation study is needed to show how each component of VADER addresses this problem.

Prabhudesai et al., "Aligning text-to-image diffusion models with reward backpropagation." arXiv. 2023.
Clark et al. "Directly fine-tuning diffusion models on differentiable rewards." ICLR. 2024.

**Objective of the Alignment Tuning:** The results shown in the DDPO and DPO papers use significantly larger reward query samples. In this paper, however, smaller-scale experiments were conducted to highlight the sample and computational efficiency of reward gradients. This discrepancy may have resulted in DDPO and DPO showing unusually poor results. Since the current alignment settings are data-free, with comparable test times, sample and computational efficiency are not as critical. Therefore, it would be beneficial to include comparisons with DPO and DDPO results over longer training times.

**Questions:**

**Comparison with Existing Guidance Methods:** Reward gradients are commonly used in guidance-based research within diffusion models, as seen in studies like DOODL and Universal Guidance. It would be beneficial to analyze and compare the proposed method with these approaches. If direct implementation of these methods is challenging, an alternative comparison could involve modifying the reward guidance objective to apply guidance through $\nabla_{x_t} R$ during generation, similar to the approach taken in this paper.

**Clarification in Experiments (Table 1):** It is unclear which video generation model serves as the baseline in Table 1. The experimental setup mentions the use of VideoCrafter, Open-Sora 1.2, and ModelScope. Is the baseline an average of these models, or is it based on one specific model? Further clarification on this would be helpful.

---

> ### Author Response · Authors · 2024-11-28
>
> **Q3.1: Clarity of Contribution**
>
> Please refer to the global answer.
>
>
>
> **Q 3.2: Step-by-step ablation study is needed to show how each component of VADER addresses memory overhead problem**
>
> We conduct a step-by-step ablation study to demonstrate how each component of VADER addresses the memory overhead problem. To study the contribution of each component to memory reduction, we conduct experiments on a single gpu, with batch size of 1. For this experiment, we offload the memory to the CPU main memory to prevent GPU out-of-memory error. The results are shown in the Table below.
>
> | Method                        | VRAM   | System RAM | Total RAM |
> |-------------------------------|--------|------------|-----------|
> | LoRA + Mixed Precision        | 12.1 GB | 264.2 GB   | 276.3 GB  |
> | + Subsampling Frames          | 12.1 GB | 216.8 GB   | 228.9 GB  |
> | + Truncated Backpropagation   | 12.1 GB | 57.3 GB    | 69.4 GB   |
> | + Gradient Checkpointing      | 12.1 GB | 20.4 GB    | 32.5 GB   |
>
>
>
>
> We find that the total memory is reduced by a significant amount, while using the above components.
>
>
>
>
>
>
> **Q 3.3: Include comparisons with DPO and DDPO results over longer training times**
>
> Thanks for the suggestion. We trained DPO and DDPO for longer. We found a meaningful improvement in DPO after long training, however we couldn’t see the same improvement in DDPO. We think this is because the number of gradient accumulation steps in DDPO’s implementation are very large i.e they scale linearly wrt the number of diffusion timesteps, thus the number of parameter updates done are relatively very little even with large amounts of compute. A better study on DDPO’s parameter update rate for video training, could improve its sample efficiency. The results of our experiments can be found here: https://vader-anonymous.github.io/#training-efficiency-comparison, and also in Figure 12 in the main paper.
>
>
>
>
> **Q3.4: Comparison with Existing Guidance Methods**
>
> Thanks for the suggestion, the guidance methods such as DOODL or Universal Guidance are applied for each example separately. VADER on the other hand updates the weights of the model, thus not requiring per sample adaptation at test time. In the following link: https://vader-anonymous.github.io/#doodl-vader, we compare VADER with DOODL, we find DOODL improves with more gradient update steps, however the improvement is still relatively less and scales linearly wrt number of examples we use for evaluation, thus making it difficult to be used in practice.
>
>
>
> **Q3.5 Clarification in Table 1.**
> Our experiments in Table 1 are based on ModelScope, we have clarified this in the paper. We have also conducted further quantitative experiments with other base video models, which are shown in [VBench Evaluation](https://vader-anonymous.github.io/#vbench-evaluation), [Eval Crafter](https://vader-anonymous.github.io/#evalcrafter-evaluation) [diversity test](https://vader-anonymous.github.io/#diversity-test).

---

> > ### Comment · Reviewer_wLqi · 2024-11-29
> >
> > I would like to express my gratitude to the authors for their efforts in addressing the feedback thoroughly. Their clarifications and additional experiments adequately resolved my initial concerns. The detailed explanations strengthened the validity of the methodology and results. The additional evidence presented aligns well with the claims made in the paper, enhancing its overall impact and clarity. As a result, the score has been revised from 6 to 8.

---

### Official Review · Reviewer_HJ2c · 2024-11-04

**Soundness:** 3
**Presentation:** 4
**Contribution:** 2
**Rating:** 6
**Confidence:** 3

**Summary:**

This paper addresses the problem of aligning pretrained video generation diffusion models to downstream tasks/domains using available reward functions without using fine-tuning datasets. Specifically, the authors propose updating the parameters of the diffusion model using the gradients of the target reward functions. This is motivated by the analysis provided in the paper (figure 3), showing that the feedback from reward gradients scale much more with video resolution compared to methods based on policy gradients. The authors apply their method to different reward functions, such as image aesthetics evaluation, image-text alignment, and video temporal consistency evaluation functions. The authors also incorporate some techniques to maintain efficiency when fine-tuning the retrained model. The proposed method is compared with multiple pretrained video generation models, as well as their aligned version using policy-gradient-based methods.

**Strengths:**

- The paper is very well-written and well-structured, making it easy to follow and understand.

- The provided analysis in Fig. 3, which shows the gap between the feedback from reward gradients and policy gradients for higher resolution videos, is valuable.

- The method is evaluated on multiple base video generators, showing consistent results.

- The proposed method shows better results in comparison to the policy-gradient-based baselines in terms of the evaluated metrics. This is also noticeable in the visual results.

- The proposed method has better generalizability on unseen text prompts compared to the baseline alignment methods.

**Weaknesses:**

- **Novelty**: To my understanding, the proposed method is in essence a standard fine-tuning method with the objective of maximizing task-specific discriminative/reward functions. The proposed techniques for efficiency, including LoRA, truncated back propagation, and frame subsampling are also all standard and commonly used in different areas. The amount of technical novelty is not a major concern as long as the method has significant findings. However, the behavior shown in the paper, i.e. better alignment of the diffusion model when directly optimized to maximize the target reward function, is not very surprising to me.

- **Experiments**:
    - In addition to regression in generalizability, another potential down-side of fine-tuning methods is the reduced diversity of the aligned model. Therefore, it is important to properly evaluate the diversity of the generated videos using the aligned model. For example, it would be interesting to see how diverse the videos are the same text prompt compared to the base model.
    - Additionally, I noticed no video visualizations are provided for aligned models using video reward functions. For example, in the results provided in Fig. 9 does not show much temporal motion in the generated videos. This could also relate to the previous point, where the model could sacrifice temporal variations for more consistent frames.

**Questions:**

Please see the concerns in the Weaknesses section. I am open to increasing my score, if the authors could clarify their novelty and contribution more, and address my concerns about the experiments.

---

> ### Author Response · Authors · 2024-11-28
>
> **Q2.1 Lack novelty**
>
> Please refer to the global answer.
>
>
> **Q2.2 Diversity of the generated videos using the aligned model.**
>
> Aligning a model for a specific use-case, very often results in a reduction of diversity and has been well-studied in prior works [1,2]. For instance, a pre-trained base model has the ability to generate videos of various types such as aesthetic, non-aesthetic, compressed, text-aligned or non text-aligned etc. Fine-tuning the model for alignment  forces the model to generate a specific type of videos (eg: only aesthetic and text-aligned) thus reducing its overall diversity.
>
> We find a similar result in the Table below, where we study the diversity of the models.
>
> |                      | VideoCrafter | VADER-PickScore Reward | VADER-Aesthetic and HPS Reward | VADER-Aesthetic and ViCLIP Reward |
> |----------------------|--------------|-------------------------|---------------------------------|------------------------------------|
> | **Average Variance** | **0.0037**  | 0.0026                  | 0.0023                          | 0.0031                            |
>
>
> We study diversity by generating multiple videos given a text prompt, we embed these videos via VideoMAE, and then calculate the variance of the embedding. We find that the overall diversity of the base model reduces as we align them using specific reward models. We also find that the highest diversity is achieved with VADER trained via ViCLIP+Aesthetic reward among all other  reward models.
>
> **Q2.3: No video visualizations for video reward function**
>
> Thanks for pointing this out. We have added visualizations for
>
> ViCLIP here - https://vader-anonymous.github.io/#aesthetic-and-viclip-reward
>
> V-JEPA here - https://vader-anonymous.github.io/#v-jepa-reward.
>
> To further investigate, any sacrifice in terms of temporal variations, we study temporal dynamics using VBench (https://arxiv.org/abs/2311.17982) in the table below.
>
> | Model                     | Subject Consistency | Background Consistency | Motion Smoothness | Dynamic Degree | Aesthetic Quality | Imaging Quality |
> |---------------------------|----------------------|-------------------------|-------------------|----------------|-------------------|-----------------|
> | VideoCrafter              | 0.9544              | 0.9652                 | 0.9688           | 0.5346         | 0.5752            | 0.6677          |
> | VADER-Aesthetic and HPS   | 0.9659              | 0.9713                 | **0.9734**       | 0.4741         | **0.6295**        | **0.7145**      |
> | VADER-PickScore           | **0.9668**          | **0.9727**             | 0.9726           | 0.3732         | 0.6094            | 0.6762          |
> | VADER-ViCLIP and Aesthetic| 0.9564              | 0.9662                 | 0.9714           | **0.5519**     | 0.6008            | 0.6566          |
>
> We find that  VADER-ViCLIP achieves the highest dynamic-degree amongst all the baselines. Dynamic degree in Vbench is calculated by checking the dynamicness in the 2D flow predicted by RAFT (https://arxiv.org/abs/2003.12039).
>
> We also find that temporal coherence and motion quality is not hampered while using Image-based reward models such as PickScore or HPS when benchmarking using EvalCrafter (https://arxiv.org/abs/2310.11440), as shown in the Table below:
>
> | Model                           | Temporal Coherence | Motion Quality |
> |---------------------------------|--------------------|----------------|
> | VideoCrafter                    | 55.90             | 52.89          |
> | VADER-Aesthetic and HPS         | 59.65             | **55.46**          |
> | VADER-PickScore                 | **60.75**         | 54.65          |
> | VADER-Aesthetic and ViCLIP      | 57.08             | 54.25          |
>
>
> EvalCrafter calculates Motion Quality using Video Action classification models and flow prediction models such as RAFT, further it calculates Temporal Coherence  using warping error and semantic consistency.
>
>
> [1] - Understanding the Effects of RLHF on LLM Generalisation and Diversity  https://arxiv.org/abs/2310.06452
>
> [2] - One fish, two fish, but not the whole sea: Alignment reduces language models' conceptual diversity  https://arxiv.org/abs/2411.04427

---

> > ### Comment · Reviewer_HJ2c · 2024-11-29
> > **Feedback on Rebuttals**
> >
> > I thank authors for their response to my concerns and questions.
> >
> > - **Novelty**: My understanding of authors contributions, in summary, is that they apply the standard techniques for alignment of image diffusion models to the video domain for the first time, and they analyze the behavior through thorough experiments. While I find this analysis valuable, I am still not fully convinced about the novelty of the findings.
> >
> > - **Sample Diversity**: I find authors's argument on the expected reduction of diversity to a specific domain after alignment convincing. They also provide results to quantify the diversity of their method. However, it would be great to see the examples of generating multiple videos using the same prompt, to better assess the in-domain diversity of the method.

---

> > > ### Author Response · Authors · 2024-12-01
> > >
> > > **Examples for in-domain diversity**
> > >
> > > Thanks for the suggestion! Please find some examples for in-domain diversity here: https://vader-anonymous.github.io/Video_Diversity.html

---

> > > > ### Comment · Reviewer_HJ2c · 2024-12-01
> > > > **Final Feedback**
> > > >
> > > > I appreciate authors' response. I do not have major concerns about the work, and I find the experiments sufficient. The only remaining concern is the limited novelty. Therefore, I raise my score to 6 (marginally above acceptance).

---

### Official Review · Reviewer_bKeQ · 2024-11-06

**Soundness:** 2
**Presentation:** 2
**Contribution:** 1
**Rating:** 3
**Confidence:** 4

**Summary:**

This paper proposes to utilize pre-trained reward models that are learned via preferences on top of powerful vision discriminative models to adapt video diffusion models. The results across a variety of reward models and video diffusion models showcase the effectiveness of the proposed approach.

**Strengths:**

The authors propose the use of reward models aimed at enhancing the quality of generated videos. The experimental results demonstrate promising improvements in video quality, showcasing the effectiveness of the proposed approach.

**Weaknesses:**

1. Algorithm 1 represents a standard approach that utilizes reward feedback within a diffusion model framework, lacking significant innovation.
2. A critical challenge in applying reward feedback to diffusion models is the precise definition and training of the reward model. The manuscript employs certain image-based methods to establish the reward function for video generation; however, these methods may not adequately encapsulate the unique characteristics of video data. It would be beneficial if the authors developed and trained a reward model specifically tailored for video content, similar to advancements made in the field of image rewards [1], thereby contributing more meaningfully to the domain.
3. There appears to be an error in Equation (231). Could you please provide a detailed derivation to clarify this point?
4. The ordering of equations has been overlooked starting from Equation (3). Please ensure all equations are correctly sequenced for clarity and coherence.
[1] Imagereward: Learning and Evaluating Human Preferences for Text-to-Image Generation

**Questions:**

See weakness

---

> ### Author Response · Authors · 2024-11-28
>
> **Q1.1 Novelty:**
>
> Please refer to the global answer.
>
> **Q1.2 Image-based Reward function don’t adequately encapsulate the unique characteristics of video data:**
>
> We agree that the image-based reward models do not capture the temporal dynamics.
>
> We therefore, run experiments with the following video reward models:
>
> i) **Vi-CLIP** - since deadline we have included Vi-CLIP, which is a video-text model trained in a similar fashion to CLIP. We show that this improves our video-text alignment and overall temporal dynamics of VADER. We visualize the results in https://vader-anonymous.github.io/#aesthetic-and-viclip-reward.
>
> ii) **V-JEPA**-  we use the base V-JEPA model trained using SSL objective as our reward function. We show that masked autoencoding reward functions can improve the temporal consistency of the generated video. These results are shown in the https://vader-anonymous.github.io/#v-jepa-reward, and  Figure 9 in the main paper.
>
> iii) **VideoMAE** - we use VideoMAE fine-tuned for action classification as our reward model, Our results are described in Figure 10, and Table 1 in the paper.
>
> Further we ran evaluations using **VBench** (https://arxiv.org/abs/2311.17982) and found that using a video based reward function such as ViCLIP indeed improves the dynamic degree of the generated videos. The results can be found in the following link and the table below:
>
> | Model                     | Subject Consistency | Background Consistency | Motion Smoothness | Dynamic Degree | Aesthetic Quality | Imaging Quality |
> |---------------------------|----------------------|-------------------------|-------------------|----------------|-------------------|-----------------|
> | VideoCrafter              | 0.9544              | 0.9652                 | 0.9688           | 0.5346         | 0.5752            | 0.6677          |
> | VADER-Aesthetic and HPS   | 0.9659              | 0.9713                 | **0.9734**       | 0.4741         | **0.6295**        | **0.7145**      |
> | VADER-PickScore           | **0.9668**          | **0.9727**             | 0.9726           | 0.3732         | 0.6094            | 0.6762          |
> | VADER-ViCLIP and Aesthetic| 0.9564              | 0.9662                 | 0.9714           | **0.5519**     | 0.6008            | 0.6566          |
>
> As can be seen in the Table above:  the dynamic degree for VADER-ViCLIP is the highest amongst all the baselines. Dynamic degree in Vbench is calculated by checking the dynamicness in the 2D flow predicted by RAFT (https://arxiv.org/abs/2003.12039). However as can also be seen, VADER-ViCLIP doesn’t score high on other categories such as Image Quality, Aesthetic Quality.
>
> **To address the above concern:** We have trained our own video reward model by fine tuning ViCLIP. Our video reward model can better capture various dimensions of improvements such as Dynamic Degree, Image Quality, Subject Consistency and Motion Smoothness.  We are currently conducting HumanEval using our video reward model.
>
> We also find that temporal coherence and motion quality is not significantly hampered while using Image-based reward models such as PickScore or HPS when benchmarking using EvalCrafter (https://arxiv.org/abs/2310.11440), as shown in the Table below:
>
> | Model                           | Temporal Coherence | Motion Quality |
> |---------------------------------|--------------------|----------------|
> | VideoCrafter                    | 55.90             | 52.89          |
> | VADER-Aesthetic and HPS         | 59.65             | **55.46**          |
> | VADER-PickScore                 | **60.75**         | 54.65          |
> | VADER-Aesthetic and ViCLIP      | 57.08             | 54.25          |
>
>
> EvalCrafter uses Video Action classification models and flow prediction models such as RAFT to evaluate Motion Quality, further it uses warping error and semantic consistency to evaluate Temporal Coherence.
>
> **Q1.3 An error in Equation (231) and ordering missing in Equations.**
>
> Thank you for pointing this out. We have better clarified the equation you mentioned and  have updated the manuscript with numbered equations. References to equations in the text are also cross-checked and updated accordingly.

---

> > ### Author Response · Authors · 2024-12-02
> >
> > Dear Reviewer,
> >
> > Just a gentle reminder to check if you've had a chance to review our rebuttal. If there are any experiments or concerns that we could address and are still unresolved, that prevent you from accepting the paper, please let us know, as the deadline is 12 hours away.
> >
> > Thank you!

---

> ### Author Response · Authors · 2024-12-04
>
> **Q1.4) Beneficial if the authors developed and trained a reward model specifically tailored for video content**
>
>
> To investigate the benefits of training a custom video reward model, We fine-tuned PickScore reward model using custom 16,000 video-text + reward pairs. We constructed this dataset, by generating video-text pairs using VADER, we obtained the reward values using the automated metrics in VBench.
>
> As the PickScore reward model can process only image-text pairs, we adapt it to process video-text pairs. We do so by using its image encoder to obtain 1D embeddings for each frame in the video denoted by $[f_0, f_1 .. f_N]$, where $N$ is the number of frames in the video. We then use its text encoder to obtain the text embeddings denoted as $c$. We then train a 5 layer and 8 head decoder-only transformer model that takes in N+1 embedding vectors (frames + text) as input and outputs a k-dimensional vector, where k represents the number of metrics in VBench such as Subject Consistency,Background Consistency etc. We train this model via supervised learning using MSE loss to predict the normalized scores of VBench metrics. We freeze the PickScore Image and Text encoders, and only train our custom transformer from scratch. We visualize the training curves for the reward models here -
> https://vader-anonymous.github.io/#vbench-reward-model-train. Note that we train this reward model as the VBench metric is not directly differentiable and thus cannot be directly used in VADER. Further even if made differentiable somehow, it will not be feasible  computationally as they use a lot of models.
>
> We then use this VBench-distilled reward model discussed above to finetune VADER-Aesthetic+HPS, we plot the reward curve for fine tuning here - https://vader-anonymous.github.io/#vader-vbench-loss
>
> We find that the VBench metrics (EvalCrafter prompts) when using unseen EvalCrafter prompts improves as a result. We find that we get a better Weighted Average Score (across all metrics), over all the baselines.
>
>
> | Model                          | Subject Consistency | Background Consistency | Motion Smoothness | Dynamic Degree | Aesthetic Quality | Imaging Quality | Weighted Average |
> |--------------------------------|---------------------|-------------------------|-------------------|----------------|-------------------|-----------------|------------------|
> | VideoCrafter                   | 0.9544              | 0.9652                  | 0.9688            | 0.5346         | 0.5752            | 0.6677          | 0.7997           |
> | T2V Turbo (4 Steps)            | 0.9639              | 0.9656                  | 0.9562            | 0.4771         | 0.6183            | **0.7266**          | 0.8126           |
> | T2V Turbo (8 Steps)            | **0.9735**          | **0.9736**              | 0.9572            | 0.3686         | 0.6265            | 0.7168          | 0.8058           |
> | VADER-Aesthetic and HPS        | 0.9659              | 0.9713                  | **0.9734**        | 0.4741         | 0.6295        | 0.7145          | 0.8167       |
> | VADER-PickScore                | 0.9668              | 0.9727                  | 0.9726            | 0.3732         | 0.6094            | 0.6762          | 0.7971           |
> | VADER-Aesthetic and ViCLIP     | 0.9564              | 0.9662                  | 0.9714            | **0.5519**     | 0.6008            | 0.6566          | 0.8050           |
> | **VADER-VBench-distill**     | 0.9638              | 0.9678                  | 0.9691            | 0.5393     | **0.6393**            | 0.7231          | **0.8238**           |
>
>
> We visualize the qualitative results against our base-model of VADER-Aesthetic+HPS Model here -
> https://vader-anonymous.github.io/#vbench-distilled-reward-model .
> We observe that fine tuning using the VBench-Distill reward model consistently improves the dynamicness of the generated videos.

---

### Author Response · Authors · 2024-11-28

We thank the reviewers for their detailed feedback and thoughtful engagement with our work. We appreciate that several reviewers acknowledged the significance of aligning video diffusion models with reward gradients (HJ2c, wLqi, tG9V) . Reviewers also noted the clarity and quality of our presentation (HJ2c, tG9V), as well as the strong experimental results demonstrating the effectiveness of VADER across diverse reward models (HJ2c, wLqi, mSP2).

A common concern raised by reviewers is novelty. We therefore state the contributions of our work:

- We are amongst the first few works to successfully align video diffusion models using reinforcement learning. We are the first to do entirely online reinforcement learning, that is we do not use any external datasets during training. We show results using a wide range of video models, reward models and benchmarks.

- We show that using reward gradients for aligning video diffusion models is promising, and can be very resourceful (specifically in the video setting which is very compute bottlenecked). VADER with VideoCrafter as the base model requires less than 24 GPU hours  to train and can fit within 24GBs of GPU VRAM.

- Finally we perform a very dense evaluation for video reward alignment. We compare against various popular methods in the image alignment community such as Diffusion-DPO, DDPO etc. Further we show results across various benchmarks (VBench, EvalCrafter), while using various base video diffusion models such as  text to video and image to video models. Lastly we show that existing image reward models and self-supervised trained video discriminative models such as V-JEPA, VideoMAE and ViCLIP can be successfully used as reward model.

Since the submission deadline, we have added new experiments which can be summarized as follows:
- We have included results with the **ViCLIP video reward model**, which helps improve the temporal dynamics of our generated videos. (bKeQ, HJ2c, mSP2, tG9V). Experiment link - https://vader-anonymous.github.io/#aesthetic-and-viclip-reward
- We have added strong benchmarks such as **VBench** and **EvalCrafter**, where we compare temporal dynamics, consistency, subject or background correctness and various other metrics of the generated videos. (bKeQ, HJ2c, wLqi, tG9V). Experiment link - (https://vader-anonymous.github.io/#vbench-evaluation | https://vader-anonymous.github.io/#evalcrafter-evaluation)
- We have added new baselines, which include **DOODL** [1] and an **on-policy versions of DDPO and DPO** (wLqi, sDFV). Experiment link - (https://vader-anonymous.github.io/#training-efficiency-comparison | https://vader-anonymous.github.io/#doodl-vader)
- We further study how finetuning using a specific reward model affects the results on the other reward functions, thus studying the **correlation between different reward functions** (tG9V) Experiment link - https://vader-anonymous.github.io/#reward-correlation
- We also study the **diversity of generated videos in VADER** when  finetune using various reward models. (HJ2C). Experiment link - https://vader-anonymous.github.io/#diversity-test
- We **ablate various memory reduction tricks** proposed in VADER along with the **number of truncation steps** used (wLqi, mSP2).  Experiment links - (https://vader-anonymous.github.io/#memory-usage-comparison | https://vader-anonymous.github.io/#truncated-backpropagation-ablation)
- Lastly we provide **hundreds of qualitative visualization** of the generated videos from VADER and the baseline (tG9V).  Experiment link - https://vader-anonymous.github.io/Video_Gallery.html

All these experiments can be found on our updated appendix in the paper  and at https://vader-anonymous.github.io/.

We also apologize to the reviewers for a relatively late response. As we had six reviewers, it took us relatively more time to conduct all the requested experiments.

Below, we address each reviewer's concerns.

---

### Author Response · Authors · 2024-12-04

We sincerely thank the reviewers for their consistent engagement and thoughtful critique of our work.  Below we summarize the experiments we ran since our first rebuttal response:

- We **compare VADER against T2V-Turbo and various other baselines**, on Vbench and EvalCrafter. We find VADER consistently outperforms the baselines on these benchmarks. Experiment Links - (https://vader-anonymous.github.io/#vbench-evaluation-standard-prompt-suite, https://vader-anonymous.github.io/#evalcrafter-evaluation)
- We **train a custom video reward model using the automated rewards from VBench**, essentially creating a differentiable distilled VBench reward model. We then use this reward model to finetune VADER and find that this improves the dynamicness of the generated videos on unseen prompts. Experiment Links - https://vader-anonymous.github.io/#vbench-distilled-reward-model
-  We ran **qualitative visualization comparing the in-domain diversity between VADER and VideoCrafter** baselines. We do not find major reduction in diversity in VADER compared to VideoCrafter baseline. Experiment Link - https://vader-anonymous.github.io/Video_Diversity.html

We will update these results in the revised version of our paper.

---

### Meta-Review · Area_Chair_TYiK · 2024-12-15

**Metareview:**

This work presents a method to efficiently adapt large-scale video diffusion models to downstream tasks by leveraging pre-trained reward models. The results show the effectiveness of the method.

The paper is generally well-written. The results are convincing and demonstrate the effectiveness of the method.

The paper has mixed review scores: 866385.
After the rebuttal, internal discussions among reviewers and AC were initiated. Two critical issues below were identified, which led to the rejection decision.

[limited novelty] reviewers sDFV, mSP2, and HJ2c raised their concerns about novelty, though some of them gave a score of 6. Using a discriminator to directly backward the gradient is widely used in many studies and lacks significant novelty. This work applies gradient backpropagation in video diffusion models. It is incremental work, despite good performance in several benchmark tests.

[text video alignment is not addressed well using reinforcement learning] Models fine-tuned with the Aesthetic Reward Model fail to align with text, because this reward model is trained to evaluate aesthetics. As pointed out by reviewer mSP2: in Figure 8, the output does not depict a starry sky as described. Similarly, in the newly presented experimental results titled “DOODL vs. VADER,” prompts like 2 dogs and a whale, ocean adventure and Teddy bear and 3 real bears continue to exhibit this problem.

The authors are encouraged to incorporate this feedback into the revised version and resubmit the work to future conferences.

**Additional Comments On Reviewer Discussion:**

The paper has mixed review scores: 866385.
After the rebuttal, two critical issues below were identified, which led to the rejection decision.

[limited novelty] reviewers sDFV, mSP2, and HJ2c raised their concerns about novelty, though some of them gave a score of 6. Using a discriminator to directly backward the gradient is widely used in many studies and lacks significant novelty. This work applies gradient backpropagation in video diffusion models. It is incremental work, despite good performance in several benchmark tests.

[text video alignment is not addressed well using reinforcement learning] Models fine-tuned with the Aesthetic Reward Model fail to align with text, because this reward model is trained to evaluate aesthetics. As pointed out by reviewer mSP2: in Figure 8, the output does not depict a starry sky as described. Similarly, in the newly presented experimental results titled “DOODL vs. VADER,” prompts like 2 dogs and a whale, ocean adventure and Teddy bear and 3 real bears continue to exhibit this problem.

---

### Decision · Program_Chairs · 2025-01-22

Reject